# Which is Better for Learning with Noisy Labels: The Semi-supervised Method or Modeling Label Noise?

## Abstract

In real life, accurately annotating large-scale datasets is sometimes difficult. Datasets used for training deep learning models are likely to contain label noise. To make use of the dataset containing label noise, two typical methods have been proposed. One is to employ the semi-supervised method by exploiting labeled *confident examples* and unlabeled *unconfident examples*. The other one is to *model label noise* and design *statistically consistent* classifiers. A natural question remains unsolved: which one should be used for a specific real-world application? In this paper, we answer the question from the perspective of *causal data generation process*. Specifically, the semi-supervised method depends heavily on the data generation process while the modeling label-noise method is independent of the generation process. For example, for a given dataset, if it has a causal generative structure that the features cause the label, the semi-supervised method would not be helpful. When the causal structure is unknown, we provide an intuitive method to discover the causal structure for a given dataset containing label noise.

## 1 Introduction

Deep neural networks can achieve remarkable performance when accurately annotated large-scale training datasets are available. However, annotating a large number of examples accurately is often expensive and sometimes infeasible in real life. Cheap datasets which contain label errors are easy to obtain (Li et al., 2019) and have been widely used to train deep neural networks. Recent results (Han et al., 2018; Nguyen et al., 2019) show that deep neural networks can easily memorize label noise during the training, which leads to poor test performance.

To reduce the side effect of label noise, there are two major streams of methods. One stream of methods focus on getting rid of label errors. Specifically, they would first select *confident examples* (i.e., whose labels are likely to be correct), e.g., by exploiting the memorization effect of deep networks (Jiang et al., 2018). Then, by discarding the labels of *unconfident examples* (i.e., whose labels are likely to be inccorect ) and keeping their unlabeled instances, they (Li et al., 2019; 2020; Wei et al., 2020; Yao et al., 2021; Tan et al., 2021; Ciortan et al., 2021; Yao et al., 2021) would employ the semi-supervised method, e.g., mixmatch (), to achieve state-of-the-art performance. Those methods are usually based on heuristics and lack theoretical guarantee.

Another major stream is to model the label noise and then get rid of its side effects. They mainly focus on estimating the label noise *transition matrix* $\boldsymbol{T}(\boldsymbol{x})$, i.e., $\boldsymbol{T}_{ij}(\boldsymbol{x}) = P(\tilde{Y} = i | Y = j, X = \boldsymbol{x})$ representing the probability that an instance $\boldsymbol{x}$ with a clean label $Y = i$ but flips to a noisy label $\tilde{Y} = j$. The idea is that the clean class posterior distribution $P(Y|X)$ can be inferred by learning the transition matrix $\boldsymbol{T}(\boldsymbol{x})$ and noisy class posterior distribution $P(\tilde{Y}|X)$. In general, when $\boldsymbol{T}(\boldsymbol{x})$ is well estimated (or given), these methods are *statistically consistent*, i.e., they guarantee that the classifiers learned from the noisy data converge to the optimal classifiers defined on the clean data as the size of the noisy training data increases (Patrini et al., 2017; Xia et al., 2019).

It naturally raises the question that which stream of methods should be used for a specific real-world application? Answering the question is crucial for the community of learning with noisy labels. If the answer is that one stream of methods is dominating, future efforts should mainly focus on that

specific stream. However, if not, we should know their differences and which method should be used for a specific real-world application.

In this paper, from a causal perspective, we answer that none of the two streams of methods are dominating. They have advantages and disadvantages, which are closely related to the underlying data generation process. The semi-supervised methods can easily incorporate heuristics (e.g., prior knowledge) to make use of the finite training sample but they do not work if the feature is the cause of the label in the data generation process. The modeling label-noise methods are not influenced by the data generation process. They can make use of all the instances and noisy labels and can be statistically consistent but they need a large training sample to perform well.

Specifically, when the instance $X$ is a cause of the clean label $Y$, the distributions $P(X)$ and $P(Y|X)$ are disentangled (Schölkopf et al., 2012; Zhang et al., 2015), which means that $P(X)$ contains no labeling information. In other words, exploiting the unlabeled data by semi-supervised methods cannot help learn the classifier. When the clean label $Y$ is a cause of the instance $X$, the distributions of $P(X)$ and $P(Y|X)$ are entangled (Schölkopf et al., 2012; Zhang et al., 2015), then $P(X)$ generally contains some information about $P(Y|X)$. Then the semi-supervised methods are helpful. In many real-world applications, we do not know the causal structure of the data generation process. To detect that on a specific noisy dataset, we proposed an intuitive method by exploiting an asymmetric property of the two different causal structures ($X$ causes $Y$ vs $Y$ causes $X$) regarding estimating the transition matrix.

## 2 RELATED WORK

In this section, we first introduce the two major streams, i.e., the methods employing semi-supervised learning and the methods based on modeling label noise. Then we introduce the causal generation process of the noisy data.

**Method based on semi-supervised learning.** Semi-supervised learning is widely employed in learning with noisy labels. To getting rid of label errors, existing methods usually divide the dataset to confident examples and unconfident examples. Then the deep neural networks are trained on the confident examples in a supervised manner (Jiang et al., 2018; Han et al., 2018). To also make use of the unconfident examples that contain a large amount of incorrect labels, by just employing the unlabeled instances, different semi-supervised learning techniques can be employed. For example, the consistency regularization (Laine and Aila, 2016) is employed by (Englesson and Azizpour, 2021); FixMatch (Sohn et al., 2020) is employed by (Li et al., 2019); the co-Regularization is employed by (Wei et al., 2020); contrastive learning is employed by (Tan et al., 2021; Ciortan et al., 2021; Li et al., 2020; Ghosh and Lan, 2021; Yao et al., 2021; Zheltonozhskii et al., 2022). Empirically, these methods have demonstrated state-of-the-art performance.

**Method based on modeling label noise.** This family of methods mainly focuses on designing statistically consistent methods by employing the noise transition matrix $\boldsymbol{T}(\boldsymbol{x})$. Specifically, given an instance $\boldsymbol{x}$, its transition matrix $\boldsymbol{T}(\boldsymbol{x})$ reveals the transition relationship from clean labels to noisy labels of the instance., i.e.,

$$\boldsymbol{T}(\boldsymbol{x})[P(Y=1|\boldsymbol{x}),\ldots,P(Y=L|\boldsymbol{x})]^\top = [P(\tilde{Y}=1|\boldsymbol{x}),\ldots,P(\tilde{Y}=L|\boldsymbol{x})]^\top. \quad (1)$$

Let $h : \mathcal{X} \to \Delta_{C-1}$ models a class posterior distribution and $\ell_{ce}$ be the cross-entropy loss, then

$$\arg\min_h \mathbb{E}_{\boldsymbol{x},y}[\ell_{ce}(y, h(\boldsymbol{x}))] = \arg\min_h \mathbb{E}_{\boldsymbol{x},\tilde{y}}[\ell_{ce}(\tilde{y}, \boldsymbol{T}(\boldsymbol{x})h(\boldsymbol{x}))]. \quad (2)$$

The above equation shows that if $\boldsymbol{T}(\boldsymbol{x})$ is given, the minimizer of the corrected loss under the noisy distribution is the same as the minimizer of the original loss under the clean distribution (Liu and Tao, 2016; Patrini et al., 2017). In practice, $\boldsymbol{T}(\boldsymbol{x})$ usually is not given and needs to be estimated from noisy data (Xia et al., 2020; Li et al., 2021).

It is also worth mentioning that, methods focus on design robust loss functions can closely related to modeling label-noise methods. These methods usually require the noise rate to help hyperparameter selection (Zhang and Sabuncu, 2018; Liu and Guo, 2020). To calculate the noise rate, $\boldsymbol{T}(\boldsymbol{x})$ usually have to be estimated (Yao et al., 2020).

**causal generation process of noisy data.** We introduce some background knowledge about causality and describe the data generation process by the causal graph and the structural causal model (SCM) (Spirtes and Zhang, 2016). Specifically, in Fig. 1(a), we illustrate a possible data generation process when data contains instance-dependent label noise by using the causal graph which represents a flow of information and reveals causal relationships among all the variables (Glymour et al., 2019). For example, Fig. 1(a) shows that the latent clean label $Y$ is a cause of the instance $X$, and both $X$ and $Y$ are causes of $\tilde{Y}$. The generation process can also be described by a structural causal model (SCM). Specifically,

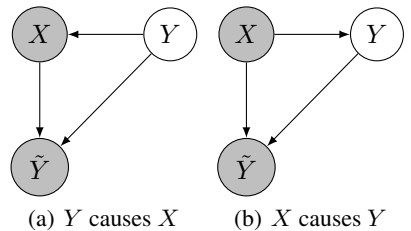

(a) $Y$ causes $X$    (b) $X$ causes $Y$

Figure 1: The change of $P(X)$ with the change of $P(Y)$ under different data generation processes.

$$Y \sim P_Y, \; U_X \sim P_{U_X}, \; X = f(Y, U_X), \; U_{\tilde{Y}} \sim P_{U_{\tilde{Y}}}, \; \tilde{Y} = g(X, Y, U_{\tilde{Y}}),$$

where $U_X$ and $U_{\tilde{Y}}$ are mutually independent exogenous random variables that are also independent of $Y$. The occurrence the exogenous variables model the random sampling process of $X$ and $\tilde{Y}$. $f$ and $g$ can be linear or non-linear functions. Each equation species a distribution of a variable conditioned on its parents (could be an empty set). Similarly, the SCM corresponding to the causal graph in Fig. 1(b) can be written as:

$$X \sim P_X, \; U_Y \sim P_{U_Y}, \; U_{\tilde{Y}} \sim P_{U_{\tilde{Y}}}, \; Y = f'(X, U_Y), \; \tilde{Y} = g(X, Y, U_{\tilde{Y}}).$$

**causal factorization and modularity.** By the conditional independence relations proposed by the Markov property (Pearl, 2000), the joint distribution $P(X, Y, \tilde{Y})$ when $Y$ causes $X$ can be factorized by following the causal direction as follows.

$$P(X, Y, \tilde{Y}) = P(Y)P(X|Y)P(\tilde{Y}|X, Y).$$

The above decomposition is called a causal decomposition. According to the *modularity property* of causal mechanisms (Schölkopf et al., 2012; Peters et al., 2017), the conditional distribution of each variable given its causes (which could be an empty set) does not inform or influence the other conditional distributions, which implies that all the distributions $P(Y)$, $P(X|Y)$ and $P(\tilde{Y}|X, Y)$ are disentangled. Similarly, when $X$ causes $Y$, the causal decomposition of $P(X, Y, \tilde{Y})$ is as follows:

$$P(X, Y, \tilde{Y}) = P(X)P(Y|X)P(\tilde{Y}|X, Y).$$

## 3 LEARNING WITH NOISY LABELS FROM A CAUSAL PERSPECTIVE

In this section, we show that the modeling label-noise method is independent of different generation processes while the semi-supervised methods depends on different generation processes. We also proposed an intuitive method to detect the causal structure by exploiting an asymmetric property regarding estimating the transition matrix.

### 3.1 THE INFLUENCE OF NOISY DATA GENERATION PROCESSES TO DIFFERENT METHODS

The modeling label-noise method is independent of different generation processes. The reason is that these methods mainly rely on estimating the transition matrix $\boldsymbol{T}(\boldsymbol{x})$, which can be estimated by exploiting the noisy class posterior $P(\tilde{Y}|X)$ learned on the noisy data (Xia et al., 2020; Li et al., 2021). It is clear that the data generation process does not influence learning $P(\tilde{Y}|X)$ and $\boldsymbol{T}(\boldsymbol{x})$.

By contrast, the semi-supervised methods are influenced by data generation processes because they rely on exploiting the unlabeled data to help learn the classifier. The helpfulness of unlabeled data depends on whether $P(X)$ contains labeling information or not. According to the causal modularity property, when $X$ causes $Y$, $P(X)$ does not contain labeling information, because $P(Y|X)$ and $P(X)$ are disentangled with each other. However, when $Y$ causes $X$, $P(X)$ should contain labeling information, because $P(X)$ and $P(Y|X)$ are entangled with each other.

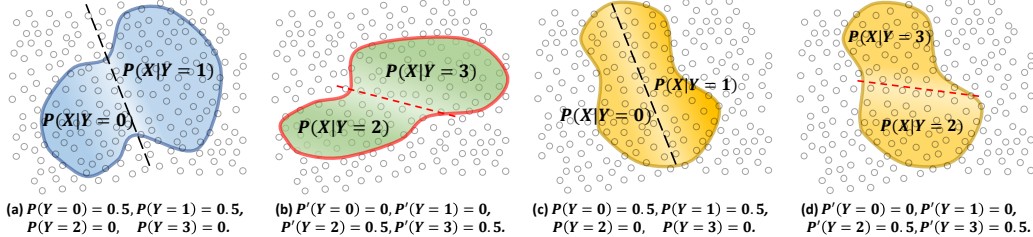

(a) $P(Y=0) = 0.5, P(Y=1) = 0.5,$ $P(Y=2) = 0,\quad P(Y=3) = 0.$

(b) $P'(Y=0) = 0, P'(Y=1) = 0,$ $P'(Y=2) = 0.5, P'(Y=3) = 0.5.$

(c) $P(Y=0) = 0.5, P(Y=1) = 0.5,$ $P(Y=2) = 0,\quad P(Y=3) = 0.$

(d) $P'(Y=0) = 0, P'(Y=1) = 0,$ $P'(Y=2) = 0.5, P'(Y=3) = 0.5.$

Figure 2: (a)-(d) illustrate the influence to $P(X)$ when $P(Y)$ changes under different data generative processes. When $Y$ causes $X$, as illustrated in (a) and (b), changing $P(Y)$ to $P'(Y)$ influences $P(X)$, then $P(X)$ contains labeling information; when $X$ causes $Y$, as illustrated in (c) and (d), changing $P(Y)$ to $P'(Y)$ does not influence $P(X)$, then $P(X)$ does not contain labeling information.

To clearly illustrate the entanglement, we will derive that, when $Y$ causes $X$, $P(Y|X)$ and $P(X)$ will change simultaneously to $P'(Y|X)$ and $P'(X)$ if we *intervene* on $Y$, i.e., change $P(Y)$ to a different distribution $P'(Y)$.

Specifically, when $P(Y)$ is changed to $P'(Y)$, $P(X|Y)$ will not be influenced because of the modularity property (Pearl, 2000). Since $P(Y)$ is changed to $P'(Y)$, and $P(X|Y)$ remains fixed, after the intervention, the joint distribution $P(X,Y) = P(Y)P(X|Y)$ will be changed to a new joint distribution $P'(X,Y) = P'(Y)P(X|Y)$. Then $P(X)$ will be changed to $P'(X) = \int_y P'(Y)P(X|Y)\mathrm{d}y$. By applying Bayes' rule, $P(Y|X) = P(Y)P(X|Y)/P(X)$ will change to a different distribution $P'(Y|X) = P'(Y)P(X|Y)/P'(X)$ unless $P'(Y)/P'(X) = P(Y)/P(X)$ which is a special case. Therefore, $P(Y|X)$ and $P(X)$ generally are entangled when $Y$ causes $X$.

To provide more intuition, we illustrate a toy example in Fig. 2. For example, as illustrated in Fig. 2(a), when $P(Y=0) = P(Y=1) = 0.5, P(Y=2) = P(Y=3) = 0$, the data is drawn from either $P(X|Y=0)$ or $P(X|Y=1)$, then $P(X) = 0.5P(X|Y=0) + 0.5P(X|Y=1)$. However, if the class prior is changed to $P'(Y=0) = P'(Y=1) = 0, P'(Y=2) = P'(Y=3) = 0.5$, as illustrated in Fig. 2(b), instead of drawing data belonging to $Y=0$ and $Y=1$, the data belonging to $Y=2$ and $Y=3$ will be drawn, and the data distribution becomes $P'(X) = 0.5P(X|Y=2) + 0.5P(X|Y=3)$. Meanwhile, the change in $P(Y)$ also leads to change in $P(Y|X)$. The changes of $P(X)$ and $P(Y|X)$ both come from changes of $P(Y)$, indicating that $P(X)$ contains information about $P(Y|X)$. Therefore the semi-supervised based methods can be useful in this case.

When feature $X$ is a cause of $Y$, intervention on $P(Y)$ will change the function $f'$ or the distribution of $U_Y$ but leave $P(X)$ unchanged. For example, from Fig. 2(c) to Fig. 2(d), the function $f'$ will be changed to output $Y=0$ or $Y=1$ instead of $Y=2$ or $Y=3$ to account for the label distribution change. The change of the selected label sets will only change the classification rules (tasks). It is clear that relabeling the sampled data points with different labels according to the new rules will not influence the distribution of the sampled data points $P(X)$, and $P(X)$ is disentangled with the different label sets. Then $P(X)$ generally does not contain information to learn clean label $Y$. Therefore the semi-supervised based methods may not work well in this case.

## 3.2 AN INTUITIVE METHOD FOR THE CAUSAL STRUCTURE DETECTION

In many real-world applications, the causal structure of the noisy data generation process is unknown. To discover the causal structure, we provide an intuitive **c**asual structure **d**etection method for learning with **n**oisy **l**abels (i.e., CDNL estimator). Our method relies on an asymmetric property of estimating flip rates under different generalization processes. Specifically, when $X$ causes $Y$, the flip rate $P(\tilde{Y}|Y')$ estimated by an unsupervised classification method usually has a large estimation error, where $Y'$ is pseudo labels estimated by the unsupervised method. However, when $Y$ causes $X$, the estimation error is small.

To be more specific, the flip rate $P(\tilde{Y}|Y')$ can be obtained by letting a clustering method estimate pseudo labels $Y'$ on all training instances. Given pseudo labels and noise labels, $P(\tilde{Y}|Y')$ then can

---

**Algorithm 1** CDNL Estimator

---

**Input:** a noisy training sample $S_{\text{tr}}$; a noisy validation sample $S_{\text{val}}$; a cluster algorithm $z$; a classification model $h$; a trainable stochastic matrix $A$

1: Optimize $h$ and $A$ via Eq. (7) to obtain $\hat{A}^* = \hat{P}(\tilde{Y}|Y^*)$ by employing the training set $S_{\text{tr}}$ and the validation set $S_{\text{val}}$;
2: Employ the cluster algorithm $z$ to estimate the cluster IDs of all instances in training set $S_{\text{tr}}$;
3: Obtain $\hat{Y}'$ of all instances from cluster IDs;
4: Calculate $\hat{P}(\tilde{Y}|Y)$ by Eq (6).

**Output:** The estimation $d(\hat{P}(\tilde{Y}|Y^*), \hat{P}(\tilde{Y}|Y'))$ via Eq. (3).

---

also be estimated. On the *causal dataset* ($X$ causes $Y$), $P(X)$ does not contain labeling information, then $Y'$ should be very different from clean label $Y$. Therefore, the estimation error of $P(\tilde{Y}|Y')$ is large. On the *anticausal dataset* ($Y$ causes $X$), $P(X)$ contains labeling information, the $Y'$ should be "close" to clean label $Y$. Therefore the estimation error of $P(\tilde{Y}|Y')$ is small.

It is worth mentioning that the performance of the proposed CDLN estimator relies on the backbone unsupervised classification method. When $Y$ causes $X$, the backbone method is expected to have reasonable classification accuracy on training instances. Thanks to the great success of the unsupervised learning methods (Likas et al., 2003; Niu et al., 2021; Ghosh and Lan, 2021; Zhou et al., 2021), some of these methods can even have compatible performance with the supervised learning on some benchmark datasets such as STL10 Coates et al. (2011) and CIFAR10 Krizhevsky et al. (2009).

Let $Y^* = \arg\max_i P(Y = i|\boldsymbol{x})$ be the Bayes label on the clean class-posterior distribution. To obtain the estimation error, we calculate the average difference between the noise rate estimated by the method based on modeling label noise and the noise rate estimated by a clustering algorithm, i.e.,

$$d(P(\tilde{Y}|Y^*), P(\tilde{Y}|Y')) = \sum_i^L \sum_j^L \frac{|P(\tilde{Y} = j|Y^* = i) - P(\tilde{Y} = j|Y' = i)|}{L^2}. \qquad (3)$$

The intuition is that given a noisy dataset, suppose that Bayes labels and pseudo labels of all instances are known and fixed, then the $P(\tilde{Y}|Y^*)$ and $P(\tilde{Y}|Y')$ are different in general unless $Y^*$ and $Y'$ are identical to each other, which as illustrated in the following theorem.

**Theorem 1.** *Let $P(Y^*|Y')$ be the transition relationship from the pseudo label $Y'$ to the Bayes label $Y^*$, then $d(P(\tilde{Y}|Y^*), P(\tilde{Y}|Y')) = 0$ either 1). for all $i \in L$, $P(Y^* = i|Y' = i) = 1$, or 2). for all $i, j \in L$ such that*

$$(1 - P(Y^* = i|Y' = i))P(\tilde{Y} = j|Y^* = i) = \sum_{k \neq i} P(Y^* = k|Y' = i)P(\tilde{Y} = j|Y^* = k), \qquad (4)$$

*where $\sum_k P(Y^* = k|Y' = i) = 1$.*

The above theorem shows to let $d(P(\tilde{Y}|Y^*), P(\tilde{Y}|Y')) = 0$, either the condition 1) or the condition 2) has to be satisfied. To satisfy the condition 1), given all examples with $Y' = i$, their Bayes labels have also be $i$, it implies that when two variables $Y^*$ and $Y'$ are identical to each other. In this case, $Y$ must cause $X$, because $P(X)$ contains labeling information, i.e., by exploiting $P(X)$, $Y^* = Y'$ can be learned. The condition 2) is a special case that requires all entries in $P(\tilde{Y}|Y^*)$ and $P(Y^*|Y')$ are carefully designed to make Eq. (4) holds, which can be hard to satisfy in general.

**Estimation of $P(\tilde{Y}|Y')$.** To estimate the flip rate $P(\tilde{Y}|Y')$, a clustering method is employed first to learn the clusters $C$. Then the clusters $C$ can be converted into the pseudo label $Y'$ by exploiting the estimated Bayes label $\hat{Y}^*$, and the average noise rate $P(\tilde{Y}|Y')$ obtained by a clustering method can be directly calculated. Be more specific, let $C = i$ denote the cluster label $i$, and let $S_{C_i} = \{\boldsymbol{x}_j\}_{j=0}^{N_{C_i}}$ denote the instance with cluster label $i$. Similarly let $S_{\hat{Y}_j^*} = \{\boldsymbol{x}_k\}_{k=0}^{N_{\hat{Y}_j^*}}$ denote the instance with estimated Bayes label $j$ by employing label-noise learning methods (Patrini et al., 2017). We assign

the pseudo labels $\hat{Y}'$ of all instances in set $S_{C_i}$ be the dominated estimated Bayes label $\hat{Y}^*$, i.e.,

$$\hat{Y}' = \arg\max_{j \in L} \frac{\sum_{x_k \in S_{\hat{Y}_j^*}} \mathbb{1}_{\{x_k \in S_{C_i}\}}}{N_{C_i}}. \tag{5}$$

Empirically, the assignment is implemented by applying Hungarian algorithm (Jonker and Volgenant, 1986). After the assignment, the pseudo labels of all training examples can be obtained. Then $P(\tilde{Y}|Y')$ can be estimated via counting on training examples, i.e.,

$$\hat{P}(\tilde{Y} = j | Y' = i) = \frac{\sum_{(\boldsymbol{x}, \tilde{y}, \hat{y}')} \mathbb{1}_{\{\hat{Y}'=i \wedge \tilde{y}=j\}}}{\sum_{(\boldsymbol{x}, \tilde{y}, \hat{y}')} \mathbb{1}_{\{\hat{Y}'=i\}}}, \tag{6}$$

where $\mathbb{1}_{\{.\}}$ is an indicator function, $(\boldsymbol{x}, \tilde{y}, \hat{y}')$ is an training example with the estimated pseudo label, and $\wedge$ represents the AND operation.

**Estimation of $P(\tilde{Y}|Y^*)$.** We estimate the average flip rate $P(\tilde{Y}|Y^*)$ in an end-to-end manner. Specifically, let $f$ be a deep classification model that outputs the estimated Bayes label in a one-hot fashion. Empirically, it can be achieved by employing Gumbel-Softmax (Jang et al., 2016). The distribution $P(\tilde{Y}|Y^*)$ is modeled by a trainable diagonally dominant column stochastic matrix $A$. Then, similar to the state-of-the-art method (Li et al., 2021), $A$ can be estimated by minimizing the empirical loss on noisy data, i.e.,

$$\{\hat{A}^*, \hat{f}\} = \arg\min_{A, f} \frac{1}{N} \sum_{\boldsymbol{x}, \tilde{y}} \ell_{ce}(\tilde{y}, Ah(\boldsymbol{x})), \quad s.t. \max_i h_i(\boldsymbol{x}) = 1. \tag{7}$$

In Section 4.1.1, we show that the estimation error of $P(\tilde{Y}|Y^*)$ by employing our method above is much smaller than employing a state-of-the-art method VolMinNet (Li et al., 2021) for both instance-dependent and instance-independent label noise. The advantage of our method to estimate $P(\tilde{Y}|Y^*)$ mainly comes from two perspectives:

1). To estimate $P(\tilde{Y}|Y^*)$ by employing existing methods, the noise transition matrix $P(\tilde{Y}|Y, X)$ has to be learned in advance, but $P(\tilde{Y}|Y, X)$ is hard to estimate in practice (Li et al., 2019; Yao et al., 2020). Our method avoids learning $P(\tilde{Y}|Y, X)$ but directly estimates the average flip rate $P(\tilde{Y}|Y^*)$. Specifically, to estimate $P(\tilde{Y}|Y^*)$ with existing methods, $P(\tilde{Y}|X)$ and $P(\tilde{Y}|Y, X)$ have to be learned first. Then both the estimated clean label $Y$ and the Bayes label $Y^*$ can be revealed by (2). After that, $P(\tilde{Y}|Y^*)$ can be estimated by using the same technique as in Eq. (6). However, because $P(\tilde{Y}|Y, X)$ usually is hard to estimate. As a result, the learned classifier (in (2)) and Bayes labels will be poorly estimated, which leads to a large estimation error of $\hat{P}(\tilde{Y}|Y^*)$. 2). We let $h$ directly estimate Bayes labels but not $\hat{P}(Y|X)$. The output complexity of $h$ reduced from a continuous distribution $\hat{P}(Y|X)$ to a discrete distribution, the learning difficulty $P(\tilde{Y}|Y^*)$ is reduced.

## 4 EXPERIMENTS

In this section, we illustrate the performance of the proposed estimator and different methods under different data generation processes with the existence of label noise.

**Baselines.** We illustrate the performance of state-of-the-art modeling label-noise methods and semi-supervised methods. The modeling label-noise methods employed are: (i) Forward (Patrini et al., 2017) which estimates the transition matrix and embeds it to the neural network; (ii) Reweighting (Liu and Tao, 2016) which gives training examples different weights according to the transition matrix by importance reweighting; (iii) T-Revision (Xia et al., 2019) which refines the learned transition matrix to improve the classification accuracy. The semi-supervised methods employed are (iv) JoCoR (Wei et al., 2020) which aims to reduce the diversity of two networks during training; (v) MoPro (Li et al., 2020) which is a contrastive learning method that achieves online label noise correction (vi) Dividemix (Li et al., 2019) which leverages the techniques FixMatch (Sohn et al., 2020) and Mixup (Zhang et al., 2018); (viii) Mixup (Zhang et al., 2018) which trains a neural network on convex combinations of pairs of examples and their labels. For all baseline methods, we follow their hyper-parameters settings mentioned in their original paper. It is worth noting that, MoPro focuses

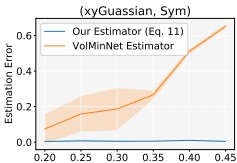 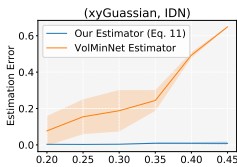 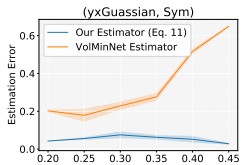 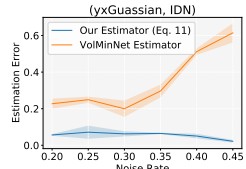

Figure 3: Estimation error of $P(\tilde{Y}|Y^*)$ on synthetic datasets with instance-independent and instance-dependent label noise. Our estimator outperforms the state-of-the-art method by a large margin.

| **XYguassian** | | Sym | | | Instance | |
|---|---|---|---|---|---|---|
| | 20% (0.196) | 30% (0.142) | 40% (0.081) | 20% (0.180) | 30 (0.127) | 40% (0.071) |
| Forward | 98.9±0.21 | 98.35±0.19 | 96.98±0.37 | 98.85±0.17 | 98.29±0.24 | 96.72±0.63 |
| Reweighting | 98.61±0.10 | **99.01**±0.12 | 96.42±1.2 | 99.54±0.23 | 99.25±0.28 | **98.37**±0.61 |
| T-Revision | **99.44**±0.12 | 98.11±0.12 | **97.08**±1.48 | **99.54**±0.23 | **99.26**±0.22 | 98.36±0.59 |
| JoCoR (SSL) | 98.05±0.03 | 97.63±0.16 | 97.11±0.19 | 98.0±0.11 | 97.65±0.21 | 97.26±0.09 |
| MoPro (SSL) | 96.75±0.67 | 95.5±1.3 | 79.76±4.95 | 95.85±0.87 | 95.26±1.78 | 78.24±6.1 |
| Dividemix (SSL) | 97.58±0.4 | 96.13±0.95 | 93.31±2.17 | 96.61±1.05 | 95.98±1.56 | 94.14±2.28 |
| Mixup (SSL) | 96.86±0.59 | 96.06±0.63 | 92.55±1.54 | 97.0±0.46 | 96.44±0.51 | 93.57±0.71 |
| **YXguassian** | | Sym | | | Instance | |
| | 20% (0.021) | 30% (0.008) | 40% (0.005) | 20% (0.023) | 30 (0.013) | 40% (0.005) |
| Forward | 86.28±0.19 | **86.04**±0.14 | **85.24**±0.41 | 86.22±0.12 | 85.98±0.23 | 85.64±0.43 |
| Reweighting | 86.23±0.14 | 85.19±0.25 | 85.13±0.68 | 86.39±0.17 | **86.04**±0.26 | 85.54±0.39 |
| T-Revision | **86.43**±0.13 | 85.2±0.12 | 85.23±0.32 | **86.4**±0.27 | 86.03±0.25 | 85.54±0.39 |
| JoCoR (SSL) | 86.14±0.08 | 85.88±0.22 | 85.23±0.53 | 86.04±0.09 | 85.86±0.28 | 85.1±0.26 |
| MoPro (SSL) | 85.17±0.71 | 83.73±1.32 | 81.11±2.35 | 85.17±0.49 | 84.4±0.54 | 82.2±1.06 |
| Dividemix (SSL) | 85.03±1.07 | 85.9±0.28 | 85.09±1.34 | 85.8±0.85 | 85.74±0.54 | **85.8**±0.36 |
| Mixup (SSL) | 85.92±0.48 | 84.3±2.34 | 82.62±2.78 | 86.2±0.22 | 85.62±0.55 | 82.08±4.57 |

Table 1: Test accuracies (%) of different methods on XYgaussian (causal) and YXgaussian (anticausal) datasets with different types of label noise. Estimations of CDNL estimator are shown in parentheses.

on image datasets, to let it work for non-image datasets, we replace the strong data augmentation for images with small Gaussian Noise, which may influence its performance.

**Datasets and noise types.** We have employed 2 synthetic datasets that are XYgaussian and YXguassian. We have also demonstrated the performance of our methods on 6 real-world datasets which are KrKp, Balancescale, Splice, waveform, MNIST, and CIFAR10. The causal datasets generated from $X$ to $Y$ are KrKp, Balancescale and Splice. The rest are anticausal datasets generated from $Y$ to $X$. We manually inject label noise into all datasets, and 20% of data is left as the validation set. Three types of noise in our experiments are employed in our experiments. (1) symmetry flipping (Sym) (Patrini et al., 2017) which randomly replaces a percentage of labels in the training data with all possible labels. (2) pair flipping (Pair) (Han et al., 2018) where labels are only replaced by similar classes. (3) instance-dependent Label Noise (IDN) (Xia et al., 2020) where different instances have different transition matrices depending on parts of instances.

**Network structure and optimization.** For a fair comparison, we implement all methods by PyTorch. All the methods are trained on Nvidia Geforce RTX 2080 GPUs. For non-image datasets, a 2-hidden-layer network with batch normalization (Ioffe and Szegedy, 2015) and dropout (0.25) (Srivastava et al., 2014) is employed as the backbone method for all baselines. We employ LeNet-5 for MNIST (LeCun, 1998) dataset and ResNet-18 (He et al., 2016) for CIFAR10 (Krizhevsky et al., 2009). To estimate $P(\tilde{Y}|Y^*)$, we use SGD to train the classification network with batch size 128, momentum 0.9, weight decay $10^{-4}$. The initial learning rate is $10^{-2}$, and it decays at 30th and 60th epochs with the rate 0.1, respectively. To get $P(\hat{Y}|Y')$, for XYguassian, yxGuassain, KrKp, Balancescale, Splice and waveform and MNIST, K-means clustering method (Likas et al., 2003) is employed; for CIFAR10, the SPICE* (Niu et al., 2021) clustering method is employed.

## 4.1 EXPERIMENTS ON SYNTHETIC DATASETS

To validate the correctness of our method, we have generated a causal dataset (from $X$ to $Y$) and an anticausal dataset (from $Y$ to $X$). For both datasets, $P(X)$ is a multivariate Gaussian mixture of $\mathcal{N}(0, \boldsymbol{I})$ and $\mathcal{N}(1, \boldsymbol{I})$ with dimension 5. For causal dataset XYguassian, the causal association $f$ and $f'$ between $X$ and $Y$ are set to be linear. The parameter of the linear function is randomly drawn

| **KrKp** | Sym | | | Instance | | |
|---|---|---|---|---|---|---|
| | 20% (0.297) | 30% (0.196) | 40% (0.070) | 20% (0.262) | 30% (0.166) | 40% (0.072) |
| Forward | 93.31±1.0 | 89.31±1.96 | 77.78±7.4 | 94.0±0.8 | 87.25±3.1 | 80.75±2.31 |
| Reweighting | 93.88±1.43 | 91.16±1.09 | 77.31±5.26 | 93.5±2.63 | 89.25±1.53 | 78.22±6.61 |
| T-Revision | **94.72**±0.62 | **91.81**±1.93 | **77.97**±5.0 | **94.5**±1.63 | **90.78**±2.35 | **79.06**±4.89 |
| JoCoR (SSL) | 93.69±0.23 | 89.53±0.84 | 67.81±2.07 | 93.44±0.71 | 87.44±2.95 | 67.75±6.51 |
| MoPro (SSL) | 89.47±1.13 | 79.47±7.03 | 65.94±2.06 | 89.31±3.82 | 79.59±6.2 | 62.62±4.78 |
| Dividemix (SSL) | 93.75±0.32 | 88.31±0.65 | 74.31±1.44 | 93.47±0.15 | 93.34±0.72 | 63.94±1.45 |
| Mixup (SSL) | 93.31±1.1 | 88.81±1.03 | 73.84±1.18 | 93.19±1.31 | 87.25±1.49 | 74.31±3.42 |
| **Balancescale** | Sym | | Pair | | Instance | |
| | 20% (0.099) | 40% (0.071) | 20% (0.113) | 40% (0.109) | 20% (0.110) | 40% (0.090) |
| Forward | 74.24±8.74 | 78.8±10.53 | 83.36±2.23 | 72.48±9.12 | 75.36±5.53 | 69.6±9.71 |
| Reweighting | 89.76±3.37 | 89.28±1.87 | **94.08**±2.41 | 79.36±15.02 | **90.72**±2.8 | **86.24**±1.38 |
| T-Revision | **92.64**±0.93 | **89.76**±3.14 | 92.32±3.97 | **81.12**±13.91 | 89.12±3.45 | 85.28±2.06 |
| JoCoR (SSL) | 76.96±3.87 | 58.08±13.43 | 72.32±10.43 | 60.16±12.88 | 73.28±4.34 | 51.2±6.13 |
| MoPro (SSL) | 84.29±2.38 | 84.13±1.81 | 84.73±3.16 | 80.79±7.93 | 86.19±2.59 | 78.1±7.28 |
| Dividemix (SSL) | 88.16±0.32 | 86.56±0.93 | 81.12±0.39 | 62.96±1.47 | 87.52±0.64 | 79.04±1.18 |
| Mixup (SSL) | 86.08±2.51 | 83.68±3.49 | 86.72±1.3 | 67.68±17.1 | 84.96±2.17 | 75.36±5.46 |
| **Splice** | Sym | | Pair | | Instance | |
| | 20% (0.136) | 40% (0.146) | 20% (0.140) | 40% (0.148) | 20% (0.151) | 40% (0.153) |
| Forward | 71.25±3.07 | 66.18±3.61 | 73.73±1.03 | 65.8±3.67 | 65.8±4.08 | 61.6±5.67 |
| Reweighting | 76.96±1.69 | 71.91±2.68 | **75.55**±1.88 | **66.68**±1.54 | 75.64±1.95 | **63.54**±7.21 |
| T-Revision | **76.99**±1.73 | **71.94**±2.68 | 75.49±2.05 | 66.61±1.5 | **75.67**±1.89 | 63.45±7.17 |
| JoCoR (SSL) | 69.81±4.61 | 63.2±1.89 | 59.37±1.44 | 57.71±3.7 | 59.66±2.44 | 55.3±5.87 |
| MoPro (SSL) | 53.6±0.19 | 53.51±0.0 | 53.51±0.0 | 53.25±0.43 | 53.79±0.38 | 52.17±3.27 |
| Dividemix (SSL) | 75.11±1.66 | 53.45±0.0 | 53.45±0.0 | 56.14±2.1 | 59.97±0.55 | 51.41±1.79 |
| Mixup (SSL) | 67.43±3.2 | 62.16±2.52 | 68.15±2.63 | 63.67±6.63 | 65.52±2.22 | 49.03±9.86 |

Table 2: Test accuracies (%) of different methods on causal datasets with different types of label noise. Estimations of CDNL estimator are shown in parentheses.

from the $\mathcal{N}(0, \boldsymbol{I})$. For YXguassian, we let the label be the mean value of the multivariate Gaussian distribution. For both datasets, we have balanced the positive and negative class priors to $0.5$, and the training sample size is 20000.

### 4.1.1 ESTIMATION ERROR OF $P(\tilde{Y}|Y^*)$

In Fig. 3, we compare the estimation error of average flip rate $P(\tilde{Y}|Y^*)$ of our CDNL estimator and the state-of-the-art method VolMinNet (Li et al., 2021), respectively. To let VolMinNet estimate $P(\tilde{Y}|Y^*)$, we first train VolMinNet with a noisy training set and select the best model by using the validation set, then the estimated clean class-posterior distribution $\hat{P}(Y|X)$ is obtained. The Bayes label $Y^*$ can be directly obtained via $\hat{P}(Y|X)$, and $P(\tilde{Y}|Y^*)$ can be estimated by using the same technique as in Eq. (6). As illustrated in Fig. 3, it shows that the estimation error of our method is close to $0$ not only on instance-independent label noise but also on instance-dependent label noise, which is much smaller than the estimated error of VolMinNet. This illustrates the advantage of CDNL estimator, which does not require learning the transition matrix and clean label for each instance, but directly estimates the average level of noise rates.

### 4.1.2 ESTIMATIONS OF CDNL ESTIMATOR AND CLASSIFICATION ACCURACY

In Tab. 1, we illustrate the estimations of CDNL estimator and the test accuracies of modeling label-noise methods and semi-supervised methods. The estimations of CDNL estimator are shown in the parentheses, and each estimation is averaged over 5 repeated experiments. The matrix $\hat{A}^* = \hat{P}(\hat{Y}|Y^*)$ estimated by our method is embedded into modeling label-noise methods.

It shows that the estimations on the anticausal dataset YXguassian are 10 times smaller than causal dataset XYguassian, which illustrates the effectiveness of our estimator. Specifically, when $X$ is a cause of $Y$ (anticausal), we expect that the difference $d(.)$ obtained by employing our estimator is large; when $Y$ is a cause of $X$, the difference $d(.)$ obtained by employing our estimator should be small. It is worth mentioning that, on both datasets, estimations of our estimator decrease with the increase of noise rates. It is because that the labels of these two datasets are binary, and $P(\tilde{Y}|Y^* = 1)$ only has one degree of freedom, i.e., $P(\tilde{Y} = 1|Y^* = 1) = 1 - P(\tilde{Y} = 0|Y^* = 1)$. Then, if the

| Waveform | Sym | | Pair | | Instance | |
|---|---|---|---|---|---|---|
| | 20% (0.138) | 40% (0.257) | 20% (0.257) | 40% (0.12) | 20% (0.099) | 40% (0.089) |
| Forward | 74.66±7.68 | 74.76±3.3 | 70.02±10.79 | 66.46±3.84 | 59.78±12.14 | 56.62±12.87 |
| Reweighting | **84.58**±1.89 | 83.92±1.38 | **83.30**±2.28 | **73.22**±4.51 | **85.02**±0.93 | **83.3**±3.02 |
| T-Revision | 84.24±1.3 | **85.70**±0.66 | 82.72±6.03 | 68.86±8.56 | 84.04±2.38 | 83.5±1.87 |
| JoCoR (SSL) | 83.44±0.83 | 60.28±1.46 | 80.64±1.29 | 57.14±4.17 | 63.84±8.8 | 54.56±4.44 |
| MoPro (SSL) | 76.62±7.16 | 76.37±7.0 | 79.55±2.32 | 58.44±7.11 | 77.36±4.04 | 65.14±5.61 |
| Dividemix (SSL) | 83.36±0.63 | 82.06±1.25 | 69.74±1.9 | 58.48±0.98 | 73.00±2.30 | 66.86±1.26 |
| Mixup (SSL) | 81.38±1.67 | 79.48±1.05 | 80.54±2.51 | 72.34±4.58 | 78.88±1.05 | 71.26±5.44 |
| **MNIST** | Sym | | Pair | | Instance | |
| | 20% (0.034) | 40% (0.038) | 20% (0.041) | 40% (0.20) | 20% (0.025) | 40% (0.026) |
| Forward | 98.75±0.08 | 97.86±0.22 | 98.84±0.10 | 94.92±0.89 | 96.87±0.15 | 90.30±0.61 |
| Reweighting | 98.71±0.11 | 98.13±0.19 | 98.54±.63 | 91.50±1.27 | 97.99±0.13 | 90.30±0.61 |
| T-Revision | 98.91±0.04 | 98.34±0.21 | 98.89±0.08 | 91.83±1.08 | 98.39±0.09 | 96.50±0.31 |
| JoCoR (SSL) | 98.06±0.13 | 96.64±0.19 | 98.01±0.19 | 96.85±0.43 | 98.62±0.06 | 96.07±0.31 |
| MoPro (SSL) | 98.51±0.92 | 95.14±1.23 | 96.79±1.04 | 94.96±1.32 | 98.53±0.52 | 96.45±1.20 |
| Dividemix (SSL) | **99.24**±0.03 | **99.21**±0.05 | **99.25**±0.03 | **98.50**±0.08 | **99.31**±0.02 | **97.75**±0.1 |
| Mixup (SSL) | 97.45±0.21 | 95.75±0.43 | 97.57±1.08 | 92.46±1.43 | 96.54±1.20 | 90.38±1.30 |
| **CIFAR10** | Sym | | Pair | | Instance | |
| | 20% (0.010) | 40% (0.009) | 20% (0.010) | 40% (0.026) | 20% (0.037) | 40% (0.042) |
| Forward | 88.21±0.48 | 78.44±0.89 | 88.21±0.48 | 77.44±6.89 | 85.29±0.38 | 74.72±3.24 |
| Reweighting | 86.77±0.40 | 83.16±0.46 | 89.60±1.01 | 77.06±6.47 | 88.72±0.41 | 84.52±2.65 |
| T-Revision | 90.33±0.52 | 84.94±2.58 | 89.75±0.41 | 80.94±2.58 | 90.46±0.13 | 85.37±3.36 |
| JoCoR (SSL) | 85.96±0.25 | 79.65±0.43 | 80.33±0.20 | 71.62±1.05 | 89.80±0.28 | 73.78±1.39 |
| MoPro (SSL) | 78.15±0.15 | 67.70±0.56 | 77.92±0.81 | 69.89±1.02 | 78.75±0.15 | 67.61±0.24 |
| Dividemix (SSL) | **95.6**±0.10 | **94.8**±1.10 | **95.72**±0.04 | **87.02** ±0.41 | **95.5**±1.17 | **94.5**±0.23 |
| Mixup (SSL) | 93.2±0.31 | 86.2±0.3 | 92.23±0.71 | 82.43±1.02 | 93.32±0.25 | 87.61±0.56 |

Table 3: Comparing test accuracies (%) of different methods on anticausal datasets with different levels and types of label noise. Estimations of CDNL estimator are shown in parentheses.

difference between $\hat{P}(\tilde{Y} = 1|Y^* = 1)$ and $\hat{P}(\tilde{Y} = 1|Y' = 1)$ is small, the difference between $\hat{P}(\tilde{Y} = 0|Y^* = 1)$ and $\hat{P}(\tilde{Y} = 0|Y' = 1)$ will also be small, the estimation will be small.

The results show that on the causal dataset XYguassian, modeling label-noise methods perform better than semi-supervised methods. It is because that $P(X)$ does not contain information of $P(Y|X)$, then semi-supervised methods may not be helpful. On the anticausal dataset YXguassian, when the sample size is large (20000), modeling label-noise methods perform slightly better than semi-supervised methods. When the complexity of anticausal datasets is high, with a limited sample size, the semi-supervised method should have better performance than modeling label-noise methods (See Tab. 3). We have also conducted an additional experiment on YXgaussian by reducing the sample size to 5000, SSL-based methods clearly outperform modeling label-noise methods under the same setting. The results are illustrated in Appendix C.

## 4.2 EXPERIMENTS ON REAL-WORLD DATASETS

We illustrate estimations of CDNL estimator, test accuracies of different methods on real-world datasets in Tab. 3 and Tab. 2. On these datasets, when the estimation of CDNL estimator is lower than 0.005, semi-supervised methods demonstrate their effectiveness, otherwise, modeling label-noise methods are more helpful to improve the robustness of learning models. It is also worth mentioning that for waveform, although it is an anticausal dataset, modeling label-noise methods have better performance than semi-supervised methods, and the estimation of CDNL estimator is also large. The reason could be that 1). $P(X)$ may not always contain information about $P(Y|X)$ even if the data generation process is from $Y$ to $X$, or 2). $P(X)$ may contain information about $P(Y|X)$, but the information can be hard to be exploited by existing methods.

## 5 CONCLUSION

In this paper, we have investigated the influence of the noisy data generation process on semi-supervised methods and modeling label-noise methods. We show that the semi-supervised methods can easily incorporate heuristics to make use of the finite training sample but their performance depends on the data generation process, while the modeling label-noise methods are independent of the generation process. In many real-world applications, the causal structure of the data generation

process is not given. Then we proposed an intuitive method by exploiting the asymmetric property of estimating flip rate under different generalization processes.

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

## A  PROOF OF THEOREM 1

In this section, we will prove Theorem 1 in our main paper.

*Proof.*
$$P(\tilde{Y} = j|Y' = i) = \sum_k P(\tilde{Y} = j|Y' = i, Y^* = k)P(Y^* = k|Y' = i)$$
$$= \sum_k P(\tilde{Y} = j|Y^* = k)P(Y^* = k|Y' = i).$$

The second equality holds because that $Y'$ is learned in an unsupervised manner that does not employ $\tilde{Y}$, then $\tilde{Y}$ does not contain any information about $\tilde{Y}$ and is independent with $\tilde{Y}$. By the above equation,

$$P(\tilde{Y} = j|Y^* = i) - P(\tilde{Y} = j|Y' = i)$$
$$= P(\tilde{Y} = j|Y^* = i) - \sum_k P(\tilde{Y} = j|Y^* = k)P(Y^* = k|Y' = i)$$
$$= (P(\tilde{Y} = j|Y^* = i) - P(\tilde{Y} = j|Y^* = i)P(Y^* = i|Y' = i)) - \sum_{k \neq i} P(\tilde{Y} = j|Y^* = k)P(Y^* = k|Y' = i)$$
$$= (1 - P(Y^* = i|Y' = i))P(\tilde{Y} = j|Y^* = i) - \sum_{k \neq i} P(Y^* = k|Y' = i)P(\tilde{Y} = j|Y^* = k).$$

For condition 2), let the difference $P(\tilde{Y} = j|Y' = i) - P(\tilde{Y} = j|Y^* = i) = 0$, then according to the above equation,

$$(1 - P(Y^* = i|Y' = i))P(\tilde{Y} = j|Y^* = i) = \sum_{k \neq i} P(Y^* = k|Y' = i)P(\tilde{Y} = j|Y^* = k).$$

For condition 1), if $P(Y^* = i|Y' = i) = 1$ for all $i$, then $(1 - P(Y^* = i|Y' = i)) = 0$ and $P(Y^* = k|Y' = i) = 0$ for all $k$. It implies that both LHS and RHS will be 0, which completes the proof.  □

## B  ESTIMATION ERROR OF $P(\tilde{Y}|Y')$

Here, we theoretically show that when $X$ causes $Y$, flip rate $P(\tilde{Y}|Y')$ estimated by an unsupervised classification method usually has a large estimation error, where $Y$ is pseudo labels estimated by the unsupervised method. However, when $Y$ causes $X$, the estimation error is usually small.

**Theorem 2.** *Let $P(\tilde{Y}|Y)$ be the transition relationship from the noisy label $\tilde{Y}$ to the clean label $Y$; let $P(\tilde{Y}|Y')$ be the transition relationship from the noisy label $\tilde{Y}$ to the pseudo label $Y'$. Then the estimation error is*

$$d(P(\tilde{Y}|Y'), P(\tilde{Y}|Y)) = \sum_i^L \sum_j^L \frac{|P(\tilde{Y} = i|Y' = j) - P(\tilde{Y} = i|Y = j)|}{L^2}$$
$$= \frac{\sum_{i,j} P(Y = j) \left| \mathbb{E}_{P(X)} \left[ \left( P(Y' = j|X)\frac{P(Y=j)}{P(Y'=j)} - P(Y = j|X) \right) \mathbb{1}_{\{\tilde{f}(X)=i\}} \right] \right|}{L^2}.$$

*Proof.* Let $\tilde{f}(x) = \arg\max_i P(\tilde{Y} = i | X = x)$ output the noisy label of every instance $x$.

$$P(\tilde{Y} = i | Y = j) = \mathbb{E}_{P(X|Y=j)}[\mathbb{1}_{\{\tilde{f}(X)=i\}}]$$
$$= \int_x \mathbb{1}_{\{\tilde{f}(x)=i\}} P(X = x | Y = j)\mathrm{d}x. \tag{8}$$

Then similarly,

$$P(\tilde{Y} = i | Y' = j) = \mathbb{E}_{P(X|Y'=j)}\left[\mathbb{1}_{\{\tilde{f}(X)=i\}}\right]$$
$$= \int_x \mathbb{1}_{\{\tilde{f}(x)=i\}} P(X = x | Y' = j)\mathrm{d}x$$
$$= \int_x \mathbb{1}_{\{\tilde{f}(x)=i\}} P(X = x | Y' = j)\frac{P(X = x | Y = j)}{P(X = x | Y = j)}\mathrm{d}x$$
$$= \mathbb{E}_{P(X|Y=j)}\left[\mathbb{1}_{\{\tilde{f}(X)=i\}}\frac{P(X | Y' = j)}{P(X | Y = j)}\right]. \tag{9}$$

The last equality is obtained by using the reweighting technique Liu and Tao (2016), which requires that $P(X|Y = j)$ and $P(X|Y' = j)$ have the same support. Then we calculate the difference $P(\tilde{Y} = i | Y' = j) - P(\tilde{Y} = i | Y = j)$ as follows.

$$P(\tilde{Y} = i | Y' = j) - P(\tilde{Y} = i | Y = j)$$
$$= \mathbb{E}_{P(X|Y=j)}\left[\mathbb{1}_{\{\tilde{f}(X)=i\}}\frac{P(X | Y' = j)}{P(X | Y = j)}\right] - \mathbb{E}_{P(X|Y=j)}[\mathbb{1}_{\{\tilde{f}(X)=i\}}]$$
$$= \int_x \mathbb{1}_{\{\tilde{f}(x)=i\}} P(X = x | Y' = j)\frac{P(X = x | Y = j)}{P(X = x | Y = j)}\mathrm{d}x - \int_x \mathbb{1}_{\{\tilde{f}(x)=i\}} P(X = x | Y = j)\mathrm{d}x$$
$$= \int_x \mathbb{1}_{\{\tilde{f}(x)=i\}} P(X = x | Y' = j)\frac{P(X = x | Y = j)}{P(X = x | Y = j)} - \mathbb{1}_{\{\tilde{f}(x)=i\}} P(X = x | Y = j)\mathrm{d}x$$
$$= \int_x \mathbb{1}_{\{\tilde{f}(x)=i\}} \left(P(X = x | Y' = j) - P(X = x | Y = j)\right)\mathrm{d}x$$
$$= \int_x \left(\frac{P(Y' = j | X = x)P(X = x)}{P(Y' = j)} - \frac{P(Y = j | X = x)P(X = x)}{P(Y = j)}\mathrm{d}x\right)\mathbb{1}_{\{\tilde{f}(X)=i\}}$$
$$= \int_x \left(\frac{P(Y' = j | X = x)P(Y = j)P(X = x)}{P(Y' = j)P(Y = j)} - \frac{P(Y = j | X = x)P(Y' = j)P(X = x)}{P(Y' = j)P(Y = j)}\right)\mathbb{1}_{\{\tilde{f}(X)=i\}}\mathrm{d}x$$
$$= \int_x \frac{P(Y' = j | X = x)P(Y = j)P(X = x) - P(Y = j | X = x)P(Y' = j)P(X = x)}{P(Y' = j)P(Y = j)}\mathbb{1}_{\{\tilde{f}(X)=i\}}\mathrm{d}x$$
$$= \int_x \frac{P(Y' = j | X = x)P(Y = j) - P(Y = j | X = x)P(Y' = j)}{P(Y' = j)P(Y = j)}P(X = x)\mathbb{1}_{\{\tilde{f}(X)=i\}}\mathrm{d}x$$
$$= \mathbb{E}_{P(X)}\left[\frac{P(Y' = j | X)P(Y = j) - P(Y = j | X)P(Y' = j)}{P(Y' = j)P(Y = j)}\right]$$
$$= P(Y = j)\mathbb{E}_{P(X)}\left[\frac{P(Y' = j | X)P(Y = j) - P(Y = j | X)P(Y' = j)}{P(Y' = j)}\mathbb{1}_{\{\tilde{f}(X)=i\}}\right]$$
$$= P(Y = j)\mathbb{E}_{P(X)}\left[\left(P(Y' = j | X)\frac{P(Y = j)}{P(Y' = j)} - P(Y = j | X)\right)\mathbb{1}_{\{\tilde{f}(X)=i\}}\right] \tag{10}$$

By using the above equation, the estimation error $d(P(\tilde{Y}|Y'), P(\tilde{Y}|Y))$ is as follows.

$$d(P(\tilde{Y}|Y'), P(\tilde{Y}|Y)) = \sum_i^L \sum_j^L \frac{|P(\tilde{Y} = i | Y' = j) - P(\tilde{Y} = i | Y = j)|}{L^2}$$
$$= \frac{\sum_{i,j} P(Y = j)\left|\mathbb{E}_{P(X)}\left[\left(P(Y' = j | X)\frac{P(Y=j)}{P(Y'=j)} - P(Y = j | X)\right)\mathbb{1}_{\{\tilde{f}(X)=i\}}\right]\right|}{L^2},$$

which completes the proof. $\qquad\square$

From Theorem 2, we can find out that to let the estimation error be small, two class posterior $P(Y'|X)$ of the pseudo label and clean class posterior $P(Y|X)$ have to be similar. For example, suppose that $P(Y'|X)$ and $P(Y|X)$ are identical, $P(Y)$ also identical to $P(Y')$ because $P(Y') = \mathbb{E}[P(Y'|X)]$ and $P(Y) = \mathbb{E}[P(Y|X)]$. Then, $P(Y' = j|X = x)\frac{P(Y=j)}{P(Y'=j)} - P(Y = j|X = x) = 0$ for all $x$, and the estimation error $d(P(\tilde{Y}|Y'), P(\tilde{Y}|Y))$ is 0.

On the anticausal dataset, $P(X)$ can inform $P(Y|X)$, therefore $P(Y'|X)$ learned by exploiting $P(X)$ is close to $P(Y|X)$ Then $P(Y)$ also close to $P(Y')$ because $P(Y') = \mathbb{E}[P(Y'|X)]$ and $P(Y) = \mathbb{E}[P(Y|X)]$. Therefore $P(Y' = j|X)\frac{P(Y=j)}{P(Y'=j)} - P(Y = j|X)$ is small. The estimation error is small. On the causal dataset, $P(X)$ can not inform $P(Y|X)$, then $P(Y'|X)$ and $P(Y|X)$ should have a large difference. Then $P(Y' = j|X)\frac{P(Y=j)}{P(Y'=j)} - P(Y = j|X)$ is usually larger than the case of $Y$ causes $X$. Then the estimation error on the causal dataset is larger than the anticausal dataset.

## C  EXPERIMENTS ON YXGAUSSIAN WITH SAMPLE SIZE 5000

We conduct additional experiments on anticausal YXgaussian dataset by reducing the training sample size to 5000. The other settings are the same as Section in the main paper. SSL-based methods clearly outperform modeling label-noise methods. The results are illustrated in Tab. 4.

| YXguassian | Sym | | | Instance | | |
|---|---|---|---|---|---|---|
| | 20% (0.037) | 30% (0.039) | 40% (0.047) | 20% (0.037) | 30 (0.036) | 40% (0.049) |
| Forward | 84.76±0.51 | 82.75±1.34 | 79.65±2.36 | 84.45±0.55 | 82.75±1.34 | 72.77±2.4 |
| Reweighting | 84.31±0.48 | 83.14±0.74 | 78.06±3.4 | 84.14±1.52 | 83.14±0.74 | 74.31±6.75 |
| T-Revision | 84.41±0.45 | 83.24±0.74 | 78.02±3.2 | 84.15±1.51 | 83.13±0.52 | 74.33±6.4 |
| JoCoR (SSL) | **85.22**±0.21 | 84.58±0.63 | 79.81±3.62 | 85.1±0.31 | 84.58±0.63 | 72.46±6.48 |
| MoPro (SSL) | 84.17±0.67 | 83.23±1.51 | 79.56±3.38 | 85.13±0.34 | 83.2±0.51 | 76.15±6.12 |
| Dividemix (SSL) | 84.73±1.21 | **84.61**±0.34 | **80.15**±4.34 | 85.6±0.76 | **85.13**±0.46 | **79.82**±4.51 |
| Mixup (SSL) | 84.6±0.41 | 83.83±2.56 | 79.43±3.63 | **85.8**±0.13 | 85.09±0.43 | 73.58±4.3 |

Table 4: Test accuracies (%) of different methods on YXgaussian datasets with sample size 5000 under different types of label noise. Estimations of CDNL estimator are shown in parentheses.

