# OpenReview forum: "Which is Better for Learning with Noisy Labels: The Semi-supervised Method or Modeling Label Noise?"
_ICLR.cc/2023/Conference — Submitted to ICLR 2023_

### Official Review · Reviewer_7ieU · 2022-10-19

**Confidence:** 3
**Correctness:** 2
**Technical Novelty And Significance:** 2
**Empirical Novelty And Significance:** 2
**Recommendation:** 3

**Clarity, Quality, Novelty And Reproducibility:**

Clarity of the technical parts could be improved, with clear definitions of different methods (groups) they are studying and how do these definitions interact with the causal data generation process. The work seems novel but relies heavily on Scho ̈lkopf et al., 2012; Peters et al., 2017. I have not reproduced the results, however the code is shared and the experiments are on public datasets with existing baselines.

**Strength And Weaknesses:**

Strengths:
S1. The problem to find when a kind of methods are better than others for learning from noisy labels is interesting and of practical relevance.
S2. The use of causal generative process to study the problem looks promising and interesting.

S3. Empirical comparison of various methods in the two groups on synthetic and real datasets.

Weaknesses:
W1. The paper makes very strong claims about two groups of methods and I have two main concerns with this: a) Looking at Table 1, I don't see significant differences b/w the two groups of the methods in XY causal and YX causal settings both. b) I don't quite see how the theory is supporting their claims either.

W2. I understand different data generation from Figure 1, however I could not understand how these differences affect different groups of methods.

W3. The algorithm 1, is too abstract and doesn't provide more details into how $h$ and $A$ are optimized and how the clusters are obtained.

**Summary Of The Paper:**

This paper studies the problem of learning with noisy labels. They broadly categorize the methods into two groups 1) that model the label noise 2) semi-supervised methods. The paper focuses on the problem that in practice it is hard to figure out which group of methods should be employed on the problem at hand. They offer a solution based on "causal data generation process". They argue that semi-supervised methods depend heavily on the data generation process while modeling label noise is independent of the data generation process. Based on this they claim that if the given dataset has a causal generative structure i.e. features causing the labels, then the semi-supervised methods would not helpful. They also give a method to estimate this causal structure from the data, as it is not available in practice.

**Summary Of The Review:**

The paper studies an interesting and relevant problem. They make strong claims about semi-supervised methods and methods that model the label noise. I am not quite convinced by the claims and the details provided to support the claims.

---

> ### Author Response · Authors · 2022-11-18
> **Response to Reviewer 7ieU**
>
> Dear Reviewer 7ieU,
>
> For your other questions and concerns, please refer to Response to Common Questions  1 2 3  and 4.
>
>
> **Looking at Table 1, I don't see significant differences b/w the two groups of the methods in XY causal and YX causal settings both.**
>
>
> A1. We are sorry for the confusion. In our paper, the sample size of the anticausal dataset YXgaussian is 20000. For such a simple dataset with low-dimensional features, when the sample size is large, the advantage of SSL-based methods is hard to be recognized.  We have conducted an additional experiment.  By reducing the training sample size of the anticausal dataset YXgaussian to 5000, SSL-based methods clearly outperform modeling label-noise methods. The results are illustrated as follows.
>
> | YXgaussian  |                | Sym            |                |                | Instance       |                |
> |-------------|----------------|----------------|----------------|----------------|----------------|----------------|
> |            | 20\% (0.037)   | 30\% (0.039)   | 40\% (0.047)   | 20\% (0.037)   | 30 (0.036)     | 40\% (0.049)   |
> | Forward     | 84.76$\pm$0.51 | 82.75$\pm$1.34 | 79.65$\pm$2.36 | 84.45$\pm$0.55 | 82.75$\pm$1.34 | 72.77$\pm$2.4  |
> | Reweighting | 84.31$\pm$0.48 | 83.14$\pm$0.74 | 78.06$\pm$3.4  | 84.14$\pm$1.52 | 83.14$\pm$0.74 | 74.31$\pm$6.75 |
> | T-Revision  | 84.41$\pm$0.45 | 83.24$\pm$0.74 | 78.02$\pm$3.2  | 84.15$\pm$1.51 | 83.13$\pm$0.52 | 74.33$\pm$6.4  |
> | JoCoR (SSL)      | **85.22**$\pm$0.21 | 84.58$\pm$0.63 | 79.81$\pm$3.62 | 85.1$\pm$0.31  | 84.58$\pm$0.63 | 72.46$\pm$6.48 |
> | MoPro  (SSL)      | 84.17$\pm$0.67 | 83.23$\pm$1.51 | 79.56$\pm$3.38 | 85.13$\pm$0.34 | 83.2$\pm$0.51  | 76.15$\pm$6.12  |
> | Dividemix  (SSL)   | 84.73$\pm$1.21 | **84.61**$\pm$0.34 | **80.15**$\pm$4.34 | 85.6$\pm$0.76  | **85.13**$\pm$0.46 | **79.82**$\pm$4.51  |
> | Mixup   (SSL)      | 84.6$\pm$0.41  | 83.83$\pm$2.56 | 79.43$\pm$3.63 | **85.8**$\pm$0.13  | 85.09$\pm$0.43 | 73.58$\pm$4.3  |

---

> ### Author Response · Authors · 2022-11-29
> **Are raised questions solved?**
>
> Dear Reviewer 7ieU
>
> Thanks for your valuable comments. We have tried our best to address all your mentioned concerns.
> After our response, are you still relatively negative about this work? Are there unclear explanations? We can carefully address them.
>
> Best, Paper3749 Authors

---

### Official Review · Reviewer_cjo1 · 2022-10-23

**Confidence:** 4
**Correctness:** 3
**Technical Novelty And Significance:** 3
**Empirical Novelty And Significance:** 2
**Recommendation:** 5

**Clarity, Quality, Novelty And Reproducibility:**

Overall the paper is clear and well-written. The main idea is novel at large. Reproducibility is good.


**Details Of Ethics Concerns:**

Not applicable.

**Strength And Weaknesses:**

**Strong points**

It is of significance to study advantages and disadvantages of the two popular families of methods dealing with noisy labels, and to understand conditions under which one is better than the other. The authors made a good observation from the perspective of causal data generation (namely, when the instance X is a cause of the clean label Y, modeling noise tends to be better; when Y causes X however, semi-supervised learning methods tends to perform better), and also proposed a method to discover such causal structure from noisy data. The finding is intuitive from the theoretical point of view. The proposed method was also demonstrated its effectiveness by extensive empirical experiments.

**Weak points**

The main problem is that, in the proposed method (CDNL), it is required to estimate Bayes labels $Y^*$. As Bayes labels are used to estimate $P(\tilde{Y}|Y')$ and $P(\tilde{Y}|Y^*)$, which is in turn used to calculate the metric (Eq. 3) that determines which family is better, the quality of $\hat{Y}^*$ needs to be good. However, if $\hat{Y}^*$ is estimated reasonably well from the noisy data, we will have obtained a good classifier. The why bother proceeding further? Also, it seems that some model-based or semi-supervised methods are needed to estimate Bayes labels. For example, in Page 6 Eq. 7, a diagonally dominant column stochastic matrix is used to model the label noise. The arguments seem somewhat circular.

**Summary Of The Paper:**

The paper compares two families of methods for learning with noisy labels. One is to use semi-supervised learning methods, and the other is to model label noise and design statistically consistent classifiers. The paper wants to answer the question: which one is better? From the perspective of causal data generation, the authors found that the answer depends on the underlying data generating process. Specifically, when the instance X is a cause of the clean label Y, modeling noise tends to be better. When Y causes X however, semi-supervised learning methods tends to perform better. In real-world applications, such causal structures are generally unknown. The authors proposed a method to discover the causal structure from a given dataset containing label noise.

**Summary Of The Review:**

The paper is marginally below the acceptance threshold. The decision was based on weighing the strong and weak points above.

**Major questions and comments.**

1. Figure 2 should be improved. Just to be precise, what are those contours, the shaded areas, and the straight dotted lines?

2. How do you know KrKp, Balancescale and Splice are causal datasets, while waveform, MNIST and CIFAR10 are anticausal datasets?

3. Just above Section 4.2, the paper says "When the complexity of anticausal datasets is high, with limited sample size, the semi-supervised method should have better performance than modeling label-noise methods." In Table 3, we see that the semi-supervised methods performed better in MNIST and CIFAR10. However, these two datasets have relatively large sample sizes. I don't see how the claim can be verified.


**Minor comments.**

1. Inconsistent notation. In Page 1, $T$ is row-stochastic, while in Eq. 2, it is column-stochastic.

2. Inconsistent notation. In Eq. 1, $C$ is the number of classes, while in Eq. 3, $L$ is used as the number of classes.

3. Page 4, second to the last line: "estimates" -> "estimate".

4. Page 6, in the paragraph just above Section 4: some citations should be in parentheses.

5. Table 1: for sym 30%, the best performance should be 86.04.

6. Page 9, Section 4.2, second to the last line: "from X to Y" should be "from Y to X".

---

> ### Author Response · Authors · 2022-11-18
> **Response to Reviewer cjo1**
>
> Dear Reviewer cjo1,
>
> The typos have been fixed in the revised version. Thank you very much for pointing this out. We will be grateful if you could go through Response to Common Questions  1, 2, 3 and 4. It may help better understand the major contribution of our paper.
>
>
>
>
>
> **Q1. If $Y^∗$ is estimated reasonably well from the noisy data, we will have obtained a good classifier. The why bother proceeding further?**
>
> A1. Thank you for this insightful question. The argument is not circular.
> Specifically, when estimating  $Y^∗$ by methods based on modeling label noise, the distribution $P(X)$ has not been exploited.
> However, when the dataset is anticausal, $P(X)$ can contain useful information to help learn the classifier. In this case, although a reasonably well classifier can be learned by the methods based on modeling label noise, the information contained in $P(X)$ is still valuable and should not be ignored especially for the datasets containing high-dimensional features such as image datasets.
>
> For example, although on MNIST and CIFAR, even a reasonably well classifier can be obtained by the methods based on modeling label noise, but SSL-based methods can still outperform the methods based on modeling label noise. A more concrete example is that on MNIST dataset with 20% of symmetric noise, all the methods based on modeling label noise have achieved nearly $99\\%$ accuracy which is reasonably well, however, the SSL-based method Dividemix still outperforms these methods, which achieved $99.24\\%$.
>
>
> From a theoretical point of view, the methods based on modeling label noise guarantee that given the transition relationship $P(\tilde{Y}|Y)$, the classifier learned on noisy data is the optimal classifier on clean data, if and only if given infinite amount of noisy training examples.
> However, in practice, we only have finite examples, even if the underlying $P(\tilde{Y}|Y)$ is given, the classifier learned by modeling label-noise methods is still influenced by label noise.  Therefore, we think if $P(X)$ contains some information about $P(Y|X)$, without any other prior knowledge, we should try to make use of $P(X)$ especially when data containing high-dimensional features. As understanding the data generative process is important, but given a dataset, we usually do not know the data generation process. This motivates us to design CDNL estimator for discovering whether the data is causal or anticausal.
>
> It is also worth emphasizing that our major contributions are not limited to the designing of CDNL estimator.  We are the first to link the causal theory and the application of LNL to help algorithm design. We have provided a new insight that the performance of SSL-based methods will be influenced by different generative processes, but the performance of the modeling label-noise methods is not influenced by data generation processes.
> This insight is crucial for the development of trustworthy machine-learning applications in real life. Specifically, in real-world scenarios, observable labels in the datasets can easily contain noise because a lot of datasets are mined by using cheap but imperfect methods such as querying commercial search engines, downloading social media images with tags and leveraging machine-generated labels. These methods inevitably yield samples with noisy labels.
> Therefore, our analysis based on LNL is general and can be applied to many real-world applications to help algorithm design.
>
>
>
>
> **Q2. In Table 3, we see that the semi-supervised methods performed better in MNIST and CIFAR10. However, these two datasets have relatively large sample sizes. I don't see how the claim can be verified.**
>
> A2. We are sorry for the confusion. Our major claim is that when on anticausal dataset, $P(X)$ contains relevant information about $P(Y|X)$, then SSL-based methods can help learn a classifier by exploiting the distribution of finite unlabeled data.
>
> Given the fact that MNIST and CIFAR10 are image datasets, where the dimensionality of their features is much higher than the features contained in the UCI datasets. By considering the high dimensionality of their features, the sample size on both datasets could be still relatively small. The information of $P(X)$ is still valuable and should be ignored.
>
>
>
>
>
> **Q3. How do you know KrKp, Balancescale and Splice are causal datasets, while waveform, MNIST and CIFAR10 are anticausal datasets?**
>
> A3. The causal relation of these datasets  can be found in [1,2].
>
>
> **Reference**
>
> [1]. Garg, Saurabh, et al. "A unified view of label shift estimation." Advances in Neural Information Processing Systems 33 (2020): 3290-3300.
>
> [2]. Schölkopf, Bernhard, et al. "On causal and anticausal learning." arXiv preprint arXiv:1206.6471 (2012).

---

> ### Author Response · Authors · 2022-11-29
> **Do you need more clarifications?**
>
> Dear Reviewer cjo1
>
> Thanks for your insightful and valuable comments. We have provided responses to address the concerns carefully. After our response, are you still relatively negative about this work? We are looking forward to hearing your further opinion on our vision paper. Please let us know if there are unclear explanations. We could further clarify them.
>
> Best, Paper3749 Authors

---

### Official Review · Reviewer_TTFe · 2022-10-26

**Confidence:** 2
**Correctness:** 2
**Technical Novelty And Significance:** 3
**Empirical Novelty And Significance:** Not applicable
**Recommendation:** 3

**Clarity, Quality, Novelty And Reproducibility:**

The paper does not introduce the influence of the data generative processes clearly (in particular, it is unclear how the outlines in Figure 2 actually represent probabilities).

Additionally, there are a relatively large number of typographical issues. Some examples:
  - inccorect on page 1
  - mixmatch on page 1 has a broken citation
  - Gaussian is written as guassian multiple times in the paper

**Details Of Ethics Concerns:**

No ethic concerns

**Strength And Weaknesses:**

The paper studies an interesting problem of comparing two major lines of work, but the conclusions are not strongly supported. There is little theoretical justification in the paper, and the experiments do not closely match the conclusions drawn in the paper.

Major Comments:

1) There are many probability distributions that can be modeled under either (a) or (b) of Figure 1; it's unclear how the causal structure be directly used to determine which set of methods would perform better

2) A core claim is that the estimation error of P(\tilde Y | Y') is more accurate, but this does not seem to have any justification.

3) There appears to be mismatches in the experimental results and the main conclusions of the paper.

Even in the datasets where semi-supervised methods are supposed to generally perform well, it seems common that only one of the methods actually performs well, while the others still have poor performance.

The tables also have incorrect numbers bolded in a number of columns (at the very least, JoCor outperformed T-Revision in XYgaussian Sym 30%, Forward outperformed Dividemix in YXgaussian Sym 30%, and Dividemix outperformed T-revision in KrKp Instance 30%), which falsely suggests that the trends are stronger than they actually are.

**Summary Of The Paper:**

This paper studies two methods (modeling label noise and semi-supervised methods) for handling datasets with noisy labels, and the causal structure strongly influences which of the methods perform better. The paper additionally proposes a method for finding the causal structure.

**Summary Of The Review:**

The paper studies an interesting problem, but there is no theoretical justification, and the experiments to not strongly support the conclusions drawn by the paper.

---

> ### Author Response · Authors · 2022-11-18
> **Response to Reviewer TTFe**
>
> Dear Reviewer TTFe,
>
> For your other questions and concerns, please refer to Response to Common Questions 1, 2 and 3. The typos have been fixed in the revised version. Thank you very much for pointing this out.
>
>
> **Q1. It is not clear what the differences are in the rows of Table 1, in both major rows, the dataset is XYgaussian**
>
>  In Table 1, the first major rows contain the results of  XYgaussian,  and the second major rows contain the results of YXgaussian.
>
>
> It is worth emphasizing that on anticausal YXgaussian dataset, with limited data size, SSL-based method could outperform modeling label-noise method.
> To validate this, we have conducted an additional experiment which reduces the training sample size of the anticausal dataset YXgaussian to 5000 from 20000, SSL-based methods clearly outperform modeling label-noise methods. The results are as follows.
>
> | YXgaussian  |                | Sym            |                |                | Instance       |                |
> |-------------|----------------|----------------|----------------|----------------|----------------|----------------|
> |            | 20\% (0.037)   | 30\% (0.039)   | 40\% (0.047)   | 20\% (0.037)   | 30 (0.036)     | 40\% (0.049)   |
> | Forward     | 84.76$\pm$0.51 | 82.75$\pm$1.34 | 79.65$\pm$2.36 | 84.45$\pm$0.55 | 82.75$\pm$1.34 | 72.77$\pm$2.4  |
> | Reweighting | 84.31$\pm$0.48 | 83.14$\pm$0.74 | 78.06$\pm$3.4  | 84.14$\pm$1.52 | 83.14$\pm$0.74 | 74.31$\pm$6.75 |
> | T-Revision  | 84.41$\pm$0.45 | 83.24$\pm$0.74 | 78.02$\pm$3.2  | 84.15$\pm$1.51 | 83.13$\pm$0.52 | 74.33$\pm$6.4  |
> | JoCoR (SSL)      | **85.22**$\pm$0.21 | 84.58$\pm$0.63 | 79.81$\pm$3.62 | 85.1$\pm$0.31  | 84.58$\pm$0.63 | 72.46$\pm$6.48 |
> | MoPro  (SSL)      | 84.17$\pm$0.67 | 83.23$\pm$1.51 | 79.56$\pm$3.38 | 85.13$\pm$0.34 | 83.2$\pm$0.51  | 76.15$\pm$6.12  |
> | Dividemix  (SSL)   | 84.73$\pm$1.21 | **84.61**$\pm$0.34 | **80.15**$\pm$4.34 | 85.6$\pm$0.76  | **85.13**$\pm$0.46 | **79.82**$\pm$4.51  |
> | Mixup   (SSL)      | 84.6$\pm$0.41  | 83.83$\pm$2.56 | 79.43$\pm$3.63 | **85.8**$\pm$0.13  | 85.09$\pm$0.43 | 73.58$\pm$4.3  |

---

> ### Author Response · Authors · 2022-11-29
> **Are there unclear explanations?**
>
> Dear reviewer TTFe,
>
> Thank you for your valuable comments. We have tried our best to address your mentioned concerns.  Are there unclear explanations here? We could further clarify them.
>
> Best, Paper3749 Authors

---

### Official Review · Reviewer_ZHc2 · 2022-10-28

**Confidence:** 3
**Clarity, Quality, Novelty And Reproducibility:** Please see the Strengths and Weakness…
**Correctness:** 3
**Technical Novelty And Significance:** 2
**Empirical Novelty And Significance:** 1
**Recommendation:** 3

**Strength And Weaknesses:**

I'd like to start by expressing that I'm not totally confident in my judgement, therefore I'm interested in the author's reponse to these statements/questions.

Strengths

-The work addresses a significant issue in dealing with training models on large datasets with noisy labels.
- The authors suggest an intriguing method termed CDNL, which computes the flip rate using a clustering method to assess causality structure.
- The proposed method appears to outperform existing methods in computing the average flip rate, which aids in detecting the dataset's causality structure.

Weaknesses
- It is not clear what the contributions and conclusions of this work are. It would help if they are explicitly and concisely listed at the end of the introduction and the abstract. Currently they seem scattered over the article and cluttered. For instance, if the proposed estimator outperforms VolMinNet in error estimation, then that should be mentioned earlier on as part of the contributions.

- I am missing how the works described in the related work related to the current work. A sentence or two about the limitations of existing methods and how the current method addresses them at the end of each paragraph would make things clearer. For example, the paragraph on "Method based on semi-supervised learning" could have an extra such sentence(s).  Right now it is not clear how this work contributes to the literature in comparison to existing works.

- I am having a hard time understanding Figure 2, could you clarify how that figure shows that semi-supervised learning does not work when P(X) does not contain information to learn clean label Y? Further, the caption in Figure 2 repeats the conditional "When Y causes X" twice with different conclusions, adding to the confusion further.

- It is not clear why the following quoted claim is true, is there a theorem that proves it? "when X causes Y , the flip rate P(Y˜ |Y ′ ) estimated by an unsupervised classification method usually has a large estimation error, where Y ′ is pseudo labels estimated by the unsupervised method. However, when Y causes X, the estimation error is small."

- this work misses large-scale real-life datasets like ImageNet which is important to reliably evaluate the methods in this work.

- More explanation and comparison needs to be done about why CDNL outperform VolMinNet. For example, is there inductive bias in CDNL that helps it learn better?

- The table is poorly presented, it is not clear which methods are semi-supervised and which are noise modeling. It would help if the authors add a column that says something like "SSL" for the semi-supervised and "NM" for the noise modeling methods

- It is not clear whether SSL is helping when there is a causal structure, so far it seems that Forward, Reweighting and T-Revision almost consistently achieve state-of-the-art except in MNIST and CIFAR10 and it is not clear why.

- It is not clear what the differences are in the rows of Table 1, in both major rows, the dataset is XYguassian. Are these differentiated by casual and anticasual?

Overall this work could use more clarification and more experiments and theorems before deriving any conclusion. The results seem a bit inconsistent to justify the use of SSL over noise-modeling and vice versa. It is not clear how this work can help us develop better methods for large datasets with noisy labels in practice

**Summary Of The Paper:**

The authors approach the challenge of huge datasets with label noise from two perspectives: a semi-supervised learning (SSL) strategy that leverages labelled and unlabeled samples, and a noise modelling (NM) approach. According to the authors, semi-supervised learning is dependent on the causal generation process, but noise modelling is not. The authors also provide a method for determining a dataset's causal structure and evaluate results between SSL and NM for different rates of causality on synthetic and real-life datasets

**Summary Of The Review:**

Please see the Strengths and Weaknesses section above.

---

> ### Author Response · Authors · 2022-11-18
> **Response to Reviewer ZHc2**
>
> Dear Reviewer ZHc2,
>
> For your other questions and concerns, please refer to Response to Common Questions 1, 2, 3 and 4.
>
>
> **Q1. It is not clear which methods in tables are semi-supervised and which are noise modeling.**
>
> A1. We are sorry for the confusion.
> The methods based on modeling label noise are  Forward, Reweighting and T-Revision.
> The SSL-based methods are JoCoR, MoPro, DivideMix and Mixup. we have added ``(SSL)’’ after the name of semi-supervised methods in the revised version.
>
> **Q2. It is not clear whether SSL is helping when there is a causal structure, it seems that Forward, Reweighting and T-Revision almost consistently achieve state-of-the-art except in MNIST and CIFAR10.**
>
> A2. For causal datasets XYguassian, KrKp, Balancescale and Splice, the methods based on modeling label noise have better performance than SSL-based methods.
>
> For anticausal datasets MNIST and CIFAR10, SSL-based methods have better performance than the methods based on modeling label noise.
>
> On the anticausal dataset YXguassian, when the sample size is large (20000), modeling label-noise methods perform slightly better than semisupervised methods
> For such a simple dataset with low-dimensional features, when the sample size is large, the advantage of SSL-based methods is hard to be recognized.  We have conducted an additional experiment by reducing the training sample size of the anticausal dataset YXgaussian to 5000, SSL-based methods clearly outperform modeling label-noise methods. The results are illustrated as follows.
>
> | YXgaussian  |                | Sym            |                |                | Instance       |                |
> |-------------|----------------|----------------|----------------|----------------|----------------|----------------|
> |            | 20\% (0.037)   | 30\% (0.039)   | 40\% (0.047)   | 20\% (0.037)   | 30 (0.036)     | 40\% (0.049)   |
> | Forward     | 84.76$\pm$0.51 | 82.75$\pm$1.34 | 79.65$\pm$2.36 | 84.45$\pm$0.55 | 82.75$\pm$1.34 | 72.77$\pm$2.4  |
> | Reweighting | 84.31$\pm$0.48 | 83.14$\pm$0.74 | 78.06$\pm$3.4  | 84.14$\pm$1.52 | 83.14$\pm$0.74 | 74.31$\pm$6.75 |
> | T-Revision  | 84.41$\pm$0.45 | 83.24$\pm$0.74 | 78.02$\pm$3.2  | 84.15$\pm$1.51 | 83.13$\pm$0.52 | 74.33$\pm$6.4  |
> | JoCoR (SSL)      | **85.22**$\pm$0.21 | 84.58$\pm$0.63 | 79.81$\pm$3.62 | 85.1$\pm$0.31  | 84.58$\pm$0.63 | 72.46$\pm$6.48 |
> | MoPro  (SSL)      | 84.17$\pm$0.67 | 83.23$\pm$1.51 | 79.56$\pm$3.38 | 85.13$\pm$0.34 | 83.2$\pm$0.51  | 76.15$\pm$6.12  |
> | Dividemix  (SSL)   | 84.73$\pm$1.21 | **84.61**$\pm$0.34 | **80.15**$\pm$4.34 | 85.6$\pm$0.76  | **85.13**$\pm$0.46 | **79.82**$\pm$4.51  |
> | Mixup   (SSL)      | 84.6$\pm$0.41  | 83.83$\pm$2.56 | 79.43$\pm$3.63 | **85.8**$\pm$0.13  | 85.09$\pm$0.43 | 73.58$\pm$4.3  |

---

> > ### Comment · Reviewer_ZHc2 · 2022-12-04
> > **Post Rebuttal**
> >
> > Unfortunately, it is still not clear whether SSL improves over noise modeling methods for anticausal datasets. In the additional results you provided here, there is strong overlap between the performance of both classes of method,  they are definitely very close within their variance. The table also has 4 SSL methods and 3 noise modeling methods so its more likely that one SSL method would outperfrom noise modeling methods randomly. If we look at some SSL methods, we see that "Mixup (SSL)" performs worse than Forward at 20%.
> >
> > Therefore, as reviewer pU2M mentioned, it might be more efficient to just try different SSL and Noise modeling methods to see which ones work better because CDNL doesn't seem to be a reliable way to find which method works.
> >
> > Further as Reviewer 7ieU there is no clear performance difference between XY causal and YX causal.
> >
> > Further, the benchmarks involve toy datasets and no large-scale datasets like mini-imagenet were used to verify these results which are necessary to see if this method is useful in real-life applications.
> >
> > I will have to lower the score to 3.

---

> ### Author Response · Authors · 2022-11-29
> **Further Discussion**
>
> Dear reviewer ZHc2,
>
> Thanks a lot for your valuable comments. We have provided responses to your major concerns. Are there unclear explanations here? We could further clarify them.
>
> Best, Paper3749 Authors

---

### Official Review · Reviewer_pU2M · 2022-11-10

**Confidence:** 4
**Correctness:** 3
**Technical Novelty And Significance:** 2
**Empirical Novelty And Significance:** 2
**Recommendation:** 5

**Clarity, Quality, Novelty And Reproducibility:**

- Several spelling mistakes across the paper, notably the consistent misspelling of causal as 'casual'
- Presentation of tables is confusing
    - Unclear at first what the rows mean
        - label noise and SSL methods should be more clearly labeled as such
        - the captions on the tables do not fully explain the table nor the main observations
    - I would produce plots of the results instead of tables since we measure how one continuous value (noise) affects another (accuracy)


**Strength And Weaknesses:**

- There are not any experiments that demonstrate how to use the CDNL estimator in order to achieve better results than simply comparing outcomes for label noise and SSL methods
    - Why would I use CDNL instead of doing both SSL and label noise methods?
    - I am not convinced that the CDNL estimator is the best method for estimating causal direction when there is a vast body of literature about how to estimate causal direction which has been ignored by this work
- The comparison with the VolMinNet is too brief: the authors would do well to outline the method of VolMinNet, what its weaknesses are, and how CDNL improves upon it


**Summary Of The Paper:**

- Assuming that there is no bi-directional causation, identify the causal direction between X and Y
- Argue that SSL is less effective in the causal direction than in the anti-causal case
- Argue that modeling label noise is agnostic to the causal direction
    - suggest a method to detect causal structure based on modeling label noise
- Introduce the CDNL estimator, which is claimed to be SOTA for $P(\tilde{Y} \mid Y^*)$
    - end to end learning of both $P(Y^* \mid X), P(\tilde{Y} \mid Y^*)$
- Demonstrate that the label noise methods outperform the SSL methods on causal datasets, while mostly matching levels of SSL methods on anti-causal datasets
    - although it is mentioned that SSL should outperform label noise methods for datasets where the ‘complexity’ is high

**Summary Of The Review:**

This paper introduces a novel method to determine the causal direction of a data generating process, when label noise is present. However, there is not enough of a literature review for me to feel comfortable that this method has not been tried in some manner before. Further, the examination of how this method can be usefully applied is poorly explained and unclear to me at present, when i compare against simply running both SSL and label noise methods in my problem of choice. Finally, the argument that anticausal and causal learning leads to different outcomes for SSL methods is not new. I would like to see more examination of the previous work on this area including methods developed (or lack thereof) to identity whether SSL is appropriate.

---

> ### Author Response · Authors · 2022-11-18
> **Response to Reviewer pU2M**
>
>
> Dear Reviewer pU2M,
>
> For your other questions and concerns, please refer to Response to Common Questions 3 and 4.
>
> **Q1. I am not convinced that the CDNL estimator is the best method for estimating causal direction.**
>
> A1. To the best of our knowledge, CDNL estimator is the first method to detect whether a dataset is causal or anticausal when data contains noisy labels.
> Specifically, existing causal discovery methods are not designed for datasets containing noisy labels. They require the instance $X$ and clean label  $Y$ to be observable to estimate causal or anticausal. When data contains noisy labels, the problem settings become that discovering the causal relation between $X$ and $Y$ when  $Y$ is latent but the noisy label $Y$ is observable.
> Then existing methods can not be directly applied because the problem settings are different.
> In contrast, CDNL estimator is specifically designed to handle the data containing noisy labels.
>
>
> **Q2. There are not any experiments that demonstrate how to use the CDNL estimator in order to achieve better results than simply comparing outcomes for label noise and SSL methods.**
>
>
> A2. The major purpose of using the CDNL estimator is to help the algorithm design when data contains noisy labels.
> If the estimation of CDNL estimator is small, then a dataset is an anticausal dataset, then it is possible to help learn $P(Y|X)$ by exploiting $P(X)$, then we could try to design a method based on different SSL techniques. For example, on anticausal datasets CIFAR10 and  MNIST, the estimation of CDNL estimator is small, then SSL-based methods can outperform the method based on modeling label noise.
>
> On the anticausal dataset YXgaussian, the estimation of CDNL estimator is small, semi-supervised methods are effective and have a similar performance to modeling label-noise methods.
>
>
> It is also worth emphasizing that our major contributions are not limited to the designing of CDNL estimator.  We are the first to link the causal theory and the application of LNL to help algorithm design. We have provided a new insight that the performance of SSL-based methods will be influenced by different generative processes, but the performance of the modeling label-noise methods is not influenced by data generation processes.  This insight is crucial for the development of trustworthy machine-learning applications in real life. Specifically, in real-world scenarios, observable labels in the datasets can easily contain noise because a lot of datasets are mined by using cheap but imperfect methods such as querying commercial search engines, downloading social media images with tags and leveraging machine-generated labels. These methods inevitably yield samples with noisy labels. Therefore, our analysis based on LNL is general and can be applied to many real-world applications to help algorithm design. We argue that, before starting the design of a real-world machine learning algorithm, it is important to understand how your data is generated. Suppose that the data generative process is known, then a lot of time, computing resources and budget could be saved in the training and testing of different algorithms.
>
>
>
>
>
>
> **Q3. Why would I use CDNL instead of doing both SSL and label noise methods?**
>
>
> A3. There are a lot of different SSL-based methods and methods modeling label noise. Each method also has to choose different backbone models (such as ResNet18, ResNext18) and hyper-parameters to work well.
> Given a real-world machine-learning dataset, it would be very expensive to properly select different hyper-parameters and try all these methods.
> However, if the data generative process is known, we can at least have a high-level idea that whether SSL-based methods or methods based on modeling label noise should be more focused on this dataset.
> This prior knowledge is important and could save a lot of time, computing resources, and budgets for the training and testing of different algorithms, which also accelerates the development of machine learning algorithms.
> However, the data generation process may not be always known in advance.
> This motivates us to propose the CDNL estimator which helps us understand how the data is generated.

---

> ### Author Response · Authors · 2022-11-29
> **Do you have remaining concerns?**
>
> Dear Reviewer pU2M,
>
> Your comments are insightful and constructive. We have provided rebuttals to address your concerns. After our response, are you still relatively negative about this work?  Do you need any more clarification? We can carefully address them.
>
> Best wishes, Paper3749 Authors

---

### Author Response · Authors · 2022-11-18
**Response to Common Question 4**


**CQ4. The comparison with the VolMinNet is too brief: the authors would do well to outline the method of VolMinNet, what its weaknesses are, and how CDNL improves upon it.**

A4. Firstly, it is worth emphasizing that the major contribution of our paper is to link the causal theory and the application of LNL to help algorithm design. Our interpretation and analysis inspire future works to first think about how to choose the algorithm design according to the data generation, which is crucial for the development of real-world machine learning applications.

We argue CDNL outperforms VolMinNet because that, to estimate $P(\tilde{Y}|Y^*)$, VolminNet needs to learn $P(\tilde{Y}|X)$ first, which is usually hard to learn. However, our method void learns $P(\tilde{Y}|Y)$ but directly learns $P(\tilde{Y}|Y^*)$.

Specifically, VolMinNet is a statistically consistent method that can learn the noise rate $P(\tilde{Y}|Y)$ and clean class posterior $P(Y|X)$ simultaneously.
Empirically, the clean class posterior $P(Y|X)$ is modeled by a neural network $f_{\theta}$ that outputs continuous values, and $P(\tilde{Y}|Y)$ is modeled by a transition matrix $T$, where the method try to jointly learn $Tf_{\theta}$ on the noisy data.

To make VolminNet estimates $P(\tilde{Y}|Y^*)$, Bayes label $Y^*$ has to be estimated first by learning both $P(\tilde{Y}|Y)$ and $f_{\theta}$ simultaneously. Since the output of  $f_{\theta}$  continuous value, the output space is relatively large in this sense. Then jointly learning $Tf_{\theta}$  could be difficult under limited training data. If $T$ and $f_{\theta}$ are not well learned, both Bayes label $Y^*$ and $P(\tilde{Y}|Y^*)$ can not be well estimated.

However, we found the to learn $P(\tilde{Y}|Y^*)$, we only need to learn the class posterior $P(Y^*|X)$ of Bayes label but not $P(Y|X)$. The class posterior $P(Y^*|X)$  of Bayes label is easier to learn than $P(Y|X)$.  Specifically, given $P(Y|X)$, $P(Y^*|X)$ can be directly inferred, however, given $P(Y^*|X)$, $P(Y|X)$ can not be inferred. It implies that $P(Y^*|X)$ is less informative than $P(Y|X)$, then $P(Y|X)$ is more difficult to learn than $P(Y^*|X)$ under the same amount of examples.
This motivates us to design a method that voids learning $P(Y|X)$ but directly learns $P(Y^*|X)$.  Empirically, instead of letting $f_{\theta}$ models $P(Y|X)$, we directly let it models $P(Y^*|X)$ by constraining the output of $f_{\theta}$ to be discrete with Gumbel-Softmax.

---

### Author Response · Authors · 2022-11-18
**Response to Common Question 3**

**CQ3. Clearly explain the contribution of this paper.**

A3. The contribution of this paper is summarized as follows.

+ From a causal perspective, by analyzing the data generative processes when data contains label noise, we found that the underlying data generative processes can have different influences on the two main streams of methods. In conclusion, the performance of SSL-based methods will be influenced by different generative processes, i.e., when $X$ causes $Y$, SSL-based methods can not leverage the unlabeled set (which is usually split from the noisy training set) to help learn $P(Y|X)$; when $Y$ causes $X$, it is possible to leverage the unlabeled set to help learn $P(Y|X)$. In contrast, the performance of the modeling label-noise methods is not influenced by data generation processes. The reason has been carefully explained in the answer of CA1.
+ We are the first to link the causal theory and the application of LNL to help algorithm design. Our interpretation and analysis inspire future works to first think about how to choose the algorithm design according to the data generation, which is crucial for the development of real-world machine learning applications. Specifically, in real-world scenarios, observable labels in the datasets can easily contain noise because a lot of datasets are mined by using cheap but imperfect methods. For example, querying commercial search engines, downloading social media images with tags, or leveraging machine-generated labels. These methods inevitably yield samples with noisy labels.  Therefore, our analysis based on LNL is general and can be applied to many real-world applications to help algorithm design.
We argue that, before starting the design of a real-world machine learning algorithm, it is important to understand how your data is generated. Suppose that the data generative process is known, we can at least have a high-level idea that whether SSL-based methods or methods based on modeling label noise should be more focused on this dataset. This prior knowledge is important and could save a lot of time, computing resources, and budgets for the training and testing of different algorithms, which also accelerates the development of machine learning algorithms.
+ We have shown that understanding the data generative process is important. However, given a dataset, we usually do not know the data generation process.  Therefore, we have proposed an intuitive method for discovering whether a dataset is causal or anticausal. This method is also backed up by theoretical analyses. To the best of our knowledge, this is the first method to whether a dataset is causal or anticausal when data contains noisy labels.

---

### Author Response · Authors · 2022-11-18
**Response to Common Question 2**

**CQ2. Theoretical proofs are required to support the claim that“when X causes Y, flip rate $P(\tilde{Y} |Y’ )$ estimated by an unsupervised classification method usually has a large estimation error, where $Y’$ is pseudo labels estimated by the unsupervised method. However, when $Y$ causes $X$, the estimation error is small.**


A2. Thank you for the advice. We have added a theorem (in Appendix B of the revised version) to support that the estimation error of $P(\tilde{Y} |Y’ )$ on the causal dataset is usually larger than that of the anticausal dataset.

**Theorem 2.** *Let $P(\tilde{Y}|Y)$ be the flip rate from the noisy label $\tilde{Y}$ to the clean label $Y$; let $P(\tilde{Y}|Y')$ be the flip rate from the noisy label $\tilde{Y}$ to the pseudo label $Y'$; let $\tilde{f}(x) = \arg\max_i P(\tilde{Y}=i|X=x)$ output the noisy label of every instance $x$. Then the estimation error is*


\begin{align}
d(P(\tilde{Y}|Y'), P(\tilde{Y}|Y))&=\sum_{i}^{L}\sum_{j}^{L} \frac{|P(\tilde{Y}=i|Y'=j) - P(\tilde{Y}=i|Y=j)|}{L^2} \\\\
    &=\frac{\sum_{i,j}P(Y=j)\left|\mathbb{E}_{ P(X)}\left[\left( P(Y'=j|X)\frac{P(Y=j)}{P(Y'=j)}-P(Y=j|X) \right) \mathbb{1}{\(\tilde{f}(X)=i\)}\right] \right|}{L^2},
\end{align}
*where $\mathbb{1}\(\tilde{f}(X)=i\)$ is the indicator function.*

From the above theorem, we can find out that to let the estimation error be small, two class posterior $P(Y'|X)$ of the pseudo label and clean class posterior $P(Y|X)$ have to be similar. For example, suppose that $P(Y'|X)$ and $P(Y|X)$ are identical, $P(Y)$ also identical to $P(Y')$ because $P(Y')=\mathbb{E}[P(Y'|X)]$ and $P(Y)=\mathbb{E}[P(Y|X)]$. Then, $P(Y'=j|X=x)\frac{P(Y=j)}{P(Y'=j)}-P(Y=j|X=x)=0$ for all $x$, and the estimation error $d(P(\tilde{Y}|Y'), P(\tilde{Y}|Y))$ is $0$.


On the anticausal dataset, $P(X)$ can inform $P(Y|X)$, therefore $P(Y'|X)$ learned by exploiting $P(X)$ is close to $P(Y|X)$ Then $P(Y)$ also close to $P(Y')$ because  $P(Y')=\mathbb{E}[P(Y'|X)]$ and $P(Y)=\mathbb{E}[P(Y|X)]$. Therefore $P(Y'=j|X)\frac{P(Y=j)}{P(Y'=j)}-P(Y=j|X)$ is small. The estimation error is small.
On the causal dataset, $P(X)$ can not inform $P(Y|X)$, then $P(Y'|X)$ and $P(Y|X)$ should have a large difference. Then $P(Y'=j|X)\frac{P(Y=j)}{P(Y'=j)}-P(Y=j|X)$ is usually larger than the case of $Y$ causes $X$. Then the estimation error on the causal dataset is larger than that of the anticausal dataset.

---

### Author Response · Authors · 2022-11-18
**Response to Common Question 1  (CQ1) [1/2]**

Dear all reviewers,


Thank you very much for your efforts in helping us review this paper. We found that there are common questions that are important and related to all reviewers. We would like to first answer these questions here.

**CQ1. There are many probability distributions that can be modeled under either causal or anticausal shown in Figure 1, it's unclear how the causal structure be directly used to determine which set of methods would perform better.**

A1. *We first explain that when analyzing from a causal perspective, the distribution of $P(X)$ is not needed to be known, but only requires that the causal data generative process is given to conclude whether a distribution can inform another distribution or not by observing the causal graph.*
This is directly from the famous modularity property of causal mechanisms, i.e., by following the causal direction, the conditional distribution of each variable (given its causes which could be an empty set) does not inform or influence the other conditional distributions.

*Now start from the simple case that given only $X$ and clean label $Y$, we analyze relations between $P(X)$ and $P(Y|X)$ under different data generative processes without label noise.*

When $X$ causes $Y$, the cause of $X$ is an empty set, and the cause of $Y$ is $X$. Given the definition of modularity property that the conditional distribution of each variable given its causes does not inform or influence the other conditional distributions, we can directly conclude that $P(X)$ can not inform $P(Y|X)$, i.e., $P(X)$ does not contain the relevant information about the predicting task.

When $Y$ causes $X$, the cause of $X$ is $Y$, and the cause of $Y$ is an empty set. Then, according to the modularity property of causal mechanisms, $P(X|Y)$ can not inform $P(X)$.
However, in this case, $P(X)$ and $P(Y|X)$ do not follow the underlying causal direction. Then they do not satisfy the modularity property anymore. Therefore, $P(X)$ can inform $P(Y|X)$. In other words, $P(X)$ generally contains the relevant information about the predicting task.

*We have explained that although we don’t know what the exact distribution of $P(X)$ is, in general, $P(X)$ contains the relevant information about the predicting task $P(Y|X)$  when $Y$ causes $X$.
Now we can explain how different causal graphs (or structures) influence semi-supervised learning (SSL).*

To make use of the unlabeled data to help learn classifiers, existing SSL relies on the condition that $P(X)$ has to contain the relevant information about the predicting task [1].
When $Y$ causes $X$, because $P(X)$ contains the relevant information of $P(Y|X)$. It is possible to help learn $P(Y|X)$ by exploiting $P(X)$ by SSL-based method.

An intuitive example is that, when $P(X)$ contains the relevant information of $P(Y|X)$, it could be possible to find some low-density areas in $P(X)$, which can separate labels. Then SSL can improve the generalization ability of a classifier by exploiting these regions in the empirical distribution of unlabeled data. This is known as low-density separation.
However, when $X$ causes $Y$, because $P(X)$  generally does not contain the relevant information of $P(Y|X)$.  It is nearly impossible to find low-density regions which can separate labels. Exploiting unlabeled data by using SSL then generally is not helpful.

*Now we can explain when data contains label noise, how the different generative processes in Fig.1 influence SSL-based methods and methods based on modeling label noise.*

In the case of learning with noisy labels (LNL), existing SSL-based methods change the problem setting from LNL to SSL by splitting a noisy training set into a labeled set (which is potentially clean) and an unlabeled set. Then different SSL techniques are employed which try to improve the performance of the classifier by exploiting the unlabeled set.
+ When $Y$ causes $X$ illustrated in Fig. 1 (a),  $P(X)$ contains the relevant information (The reason is the same as the case without label noise mentioned before, i.e., they do not follow the underlying causal direction and do not satisfy the modularity property anymore). Then SSL-based methods can use the constructed unlabeled set to help learn a classifier.
+ When $X$ causes $Y$  illustrated in Fig. 1 (b), the causal modularity suggests that both $P(\tilde{Y}|X)$ and $P(Y|X)$ will be independent of $P(X)$. $P(X)$ does not contain the relevant information, the unconfident set can not help learn a classifier in general. As a result, only a subset of the noisy training data is used for learning a classifier. It implies that SSL-based methods have the data sacrifice issue in this case.

---

> ### Author Response · Authors · 2022-11-18
> **Response to Common Question 1  [2/2]**
>
> In contrast, the consistent methods based on modeling label noise do not exploit the unlabeled set (or $P(X)$)  to learn $P(Y|X)$.
> Specifically, these methods usually first need to estimate the noise transition matrix $T(x)$, which is usually learned in a supervised manner on the whole noisy training set and does not require exploiting $P(X)$.
> Then, $P(Y|X)$ can be learned by using the estimated $T(x)$ to correct the loss on the whole noisy training set, which is also learned in a supervised manner. In these processes, only supervised information is used to help learn $P(Y|X)$ but not $P(X)$. Therefore, the performance of the methods based on modeling label noise is not influenced by the different data generative processes.  However, these methods usually require a large number of training examples to accurately estimate the transition matrix [2]. If the transition matrix is poorly estimated, the estimation error of $P(Y|X)$  will be large.
>
>
> It is worth emphasizing that we are the first to link the causal theory and the application of LNL to help algorithm design. Our interpretation and analysis inspire future works to first think about how to choose the algorithm design according to the data generation, which is crucial for the development of real-world machine learning applications.
>
> **Reference**
>
> [1]. Schölkopf, Bernhard, et al. "On causal and anticausal learning." arXiv preprint arXiv:1206.6471 (2012).
>
> [2]. Yao, Yu, et al. "Dual t: Reducing estimation error for transition matrix in label-noise learning." Advances in neural information processing systems 33 (2020): 7260-7271.

---

### Author Response · Authors · 2022-11-23
**Rolling Discussion**

Dear All Reviewers,

Thanks again for your efforts in reviewing this paper. We tried our best to address the mentioned concerns. Let us know if there are unclear explanations. We could further clarify them.

Best Regards, Paper3749 Authors

---

### Decision · Program_Chairs · 2023-01-20

**Decision:**

Reject

**Justification For Why Not Higher Score:**

There is no collective enthusiasm amongst reviewers (or myself) to champion this paper. Whilst the work has some merit, the presentation, conclusions and practical recommendations are not yet clear enough.

**Justification For Why Not Lower Score:**

N/A

**Metareview: Summary, Strengths And Weaknesses:**

The main thrust of the paper is to introduce an estimator to help determine the causal direction of the data generating process when the data labels are noisy. The paper contrasts SSL and explicit label noise modelling.  The reviewers I think found this an interesting paper, but struggled to identify clear conclusions and recommendations as to how the approach would be used in practice. The paper suffered from a lack of clarity in places and I think probably confuses the reader in places. Despite a reasonable discussion phase in which reviewers with some background in causality were present, the review scores didn't increase sufficiently to make the paper in its current form acceptable for ICLR.